# Reconstructing the Image Stitching Pipeline: Integrating Fusion and Rectangling into a Unified Inpainting Model

**Ziqi Xie[1], Weidong Zhao[1,2]\*, Xianhui Liu[1,2], Jian Zhao[1], Ning Jia[1,2]**

[1] College of Computer Science and Technology, Tongji University
[2] College of Electronics and Information Engineering, Tongji University
{xieziqi, wd, lxh, zjtju1919, jianing7072}@tongji.edu.cn

## Abstract

Deep learning-based image stitching pipelines are typically divided into three cascading stages: registration, fusion, and rectangling. Each stage requires its own network training and is tightly coupled to the others, leading to error propagation and posing significant challenges to parameter tuning and system stability. This paper proposes the Simple and Robust Stitcher (SRStitcher), which revolutionizes the image stitching pipeline by simplifying the fusion and rectangling stages into a unified inpainting model, requiring no model training or fine-tuning. We reformulate the problem definitions of the fusion and rectangling stages and demonstrate that they can be effectively integrated into an inpainting task. Furthermore, we design the weighted masks to guide the reverse process in a pre-trained large-scale diffusion model, implementing this integrated inpainting task in a single inference. Through extensive experimentation, we verify the interpretability and generalization capabilities of this unified model, demonstrating that SRStitcher outperforms state-of-the-art methods in both performance and stability. Code: https://github.com/yayoyo66/SRStitcher

## 1   Introduction

Image stitching is a fundamental problem in computer vision, which aims to obtain a larger field of view by merging multiple overlapping images [5]. As illustrated in Figure 1(a), the current deep learning-based image stitching pipeline is typically structured into three sequential stages: (1) **Registration Stage**. The first stage takes the original image pairs to estimate warping matrices, which are then used to align images. Current learning-based methods focus on designing the homography estimation networks to address the registration problem [22, 21, 7]. (2) **Fusion Stage**. The second stage merges the aligned images into a single fusion image. Present research in this domain is generally classified into reconstruction-based (recon-based) and seam-based methods. Recon-based methods [32, 36, 33] typically use the encoder-decoder networks to perform pixel-wise reconstruction of the fusion image. While seam-based methods [35, 11] focus on identifying the optimal seams to eliminate the fusion ghosting. (3) **Rectangling Stage**. The final stage transforms the irregularly shaped fusion image into a standard rectangular format. There are only a few deep learning-based studies for this stage [34, 56], and they are all supervised methods with requirements for labeling data.

Annoyingly, the cascaded structure of current image stitching pipelines poses significant challenges for training optimization and parameter tuning. Furthermore, errors from early stages tend to propagate to later stages, significantly degrading the performance of later processes. The representative

38th Conference on Neural Information Processing Systems (NeurIPS 2024).

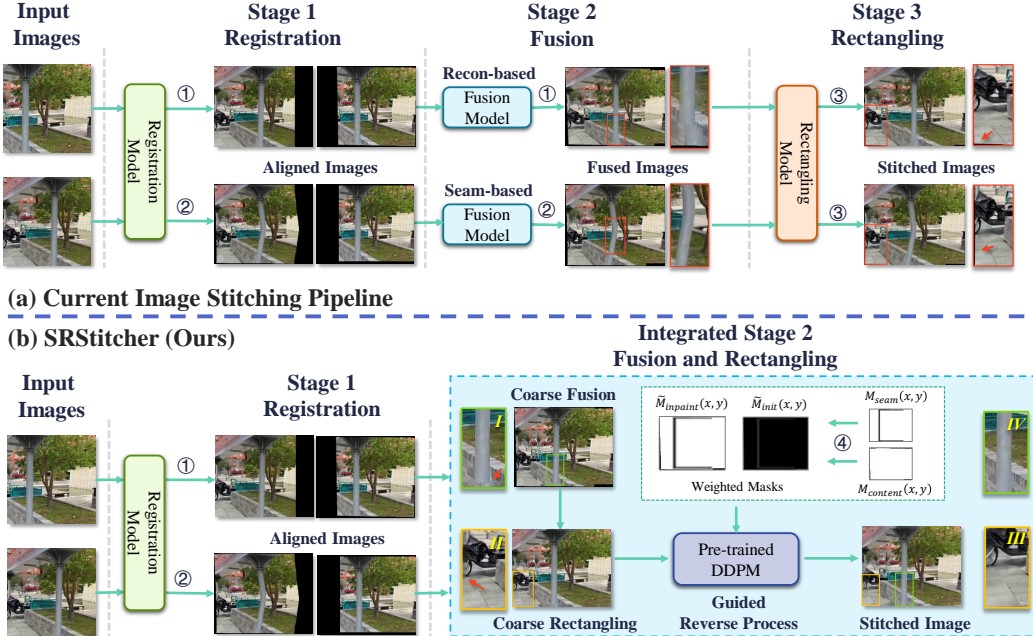

Figure 1: Comparison between existing pipeline and SRStitcher. Process ① is implemented by UDIS [33], process ② by UDIS++ [35], process ③ by DeepRectangling (DR) [34], process ④ by Eq. 10 and Eq. 11. The corresponding partial images, *I* and *IV*, illustrate how SRStitcher effectively corrects the apparent misalignment of a pillar. Similarly, the partial images *II* and *III* demonstrate how SRStitcher repairs the blurry coarse rectangling areas. SRStitcher can be applied to both UDIS and UDIS++ aligned images and get similar stitched results.

image stitching methods UDIS [33]and UDIS++ [35] both struggle to effectively fuse images with registration errors, as shown in Figure 1 ① and ② . In the rectangling stage (Figure 1 ③), the prominent rectangling method DeepRectangling (DR) [34] also fails to adequately fill gaps, leaving visible black spaces at image boundaries.

As shown in Figure 1(a), the errors originating in the registration stage persist through to the final stitched image, and the existing methods lack effective mechanisms to address these errors (detailed in Appendix A.6). The recon-based [32, 36, 33] methods are proven to inevitably introduce artifacts in the stitched image in the presence of registration errors. Furthermore, the seam-based [35, 11] methods rely on the assumption that there is a perfect seam between the aligned images. When this assumption is not met, the UDIS++ [35] forcibly distorts the images to create a *perfect seam*, resulting in distortion in the stitched image. Therefore, current fusion methods are unable to effectively handle the registration errors shown in Figure 1.

To address the error propagation problem, we identify image fusion as the key point for improvement. We reconsider the problem definition of the fusion challenge and hypothesize that ***By determining the appropriate modification region and introducing an inpainting model with strong generalization ability, the abnormal image content caused by registration error can be effectively corrected***. We propose to reformulate the fusion problem by overlaying the less distorted aligned image over the more distorted one, and inpainting the seam area between the images to correct the inappropriate image content.

Building on reframing the fusion problem, we also revisit the rectangling challenge. The core of the rectangling problem is to fill in the missing rectangling area, which is also essentially an image inpainting problem. Therefore, we question ***whether fusion and rectangling are truly distinct challenges or if they could be addressed as a unified inpainting task***. We recognize that handling fusion and rectangling tasks simultaneously is not a simple matter of determining the inpainting area. More importantly, it requires precise control of the inpainting process. Specifically, the fusion task involves the preservation of original image semantics to the greatest extent possible, while the rectangle task requires more heavy inpainting to fill the missing regions. To effectively manage

these varying demands, we introduce weighted masks to guide the reverse process in a pre-trained large-scale diffusion model. This method allows for the adjustment of inpainting intensity across different regions during the reverse process, enabling both tasks to be completed within a single inference.

The main contributions of this paper are: (1) We propose SRStitcher, which reformulates the problem definitions of the fusion and rectangling stages to construct a more streamlined and robust image stitching pipeline. (2) SRStitcher is the first to introduce the concept of inpainting to address the image fusion problem. It incorporates prior knowledge from large-scale pre-trained models into the image stitching pipeline, enhancing the robustness of image fusion against registration errors. (3) Without additional fine-tuning or supervision, SRStitcher improves the generalization of the rectangling method in the zero-shot scenario, opening up new possibilities for unsupervised image rectangling research. (4) We conduct extensive experiments to verify the interpretability and generalization of the proposed unified model. The results show that SRStitcher significantly outperforms the state-of-the-art methods in both quantitative and qualitative evaluations.

## 2   Background

**Registration parameterization**. The goal of the registration stage is to obtain the aligned images based on a transformation matrix. Given inputs $I_l(x, y), I_r(x, y) \in \mathbb{R}^{H \times W}$, where $x$ and $y$ represent the pixel coordinates, $H, W$ are the height and width, respectively. And $\mathcal{H}$ donates a $3 \times 3$ homography matrix between $I_l(x, y)$ and $I_r(x, y)$, which maps the input images to an uniform plane. To clarify the process of image registration, take the example of the four vertex coordinates $(x_k, y_k), k \in \{1, 2, 3, 4\}$ of the input image. The new image stitching-domain $\mathbb{R}^{H^* \times W^*}$ can be obtained by Eq. 1.

$$W^* = \max_{k \in (1,2,3,4)} \{x_k^w, x_k^l\} - \min_{k \in (1,2,3,4)} \{x_k^w, x_k^l\}, \tag{1a}$$

$$H^* = \max_{k \in (1,2,3,4)} \{y_k^w, y_k^l\} - \min_{k \in (1,2,3,4)} \{y_k^w, y_k^l\}, \tag{1b}$$

where, $(x_k^w, y_k^w) = \mathcal{H} \times [x_k^r, y_k^r, 1]^T$. Then, the input images are mapped into this new image stitching-domain by warping operation $\varphi(\cdot)$ to get the aligned images $I_{wl}(x, y), I_{wr}(x, y) \in \mathbb{R}^{H^* \times W^*}$, as shown in Eq. 2.

$$I_{wl}(x, y), I_{wr}(x, y) = \varphi(I_l(x, y), \mathtt{I}), \varphi(I_r(x, y), \mathcal{H}), \tag{2}$$

where, $\mathtt{I}$ is an identity matrix. The masks $M_{wl}(x, y), M_{wr}(x, y)$ corresponding to the aligned images can be obtained in a similar way by Eq. 2, except that the inputs $I_l(x, y), I_r(x, y)$ are replaced with two all-one matrixes. The specific design of $\varphi(\cdot)$ vary slightly among different stitching methods [33, 35], but the aligned image generation of these methods all follows the architecture of Eq. 2.

**Diffusion model**. The proposed work is based on the Diffusion Model [16]. Since our method does not include the forward process, we only briefly introduce the reverse process. Suppose the $\mathbf{x}_1, ..., \mathbf{x}_T$ are latents of the same dimensionality as the $\mathbf{x}_0 \sim q(\mathbf{x}_0)$, where $q(\cdot)$ is the a Gaussian Markov chain forward process with $T$ steps. And, $\mathbf{x}_0 = \mathcal{E}(I_0(x, y))$, where $\mathcal{E}(\cdot)$ is the image encoder and $I_0(x, y)$ is the input image. The joint distribution of $t$-th inversion step is defined as Eq. 3.

$$p_\theta(\mathbf{x}_{t-1} | \mathbf{x}_t) = \mathcal{N}(\mathbf{x}_{t-1}; \mu_\theta(\mathbf{x}_t, t), \Sigma_\theta(\mathbf{x}_t, t)), t \in (1, T), \tag{3}$$

where, $\mu_\theta(\mathbf{x}_t, t)$ and $\Sigma_\theta(\mathbf{x}_t, t)$ are parameters of the Gaussian Markov chain in $t$-th inversion step.

## 3   Methodology

### 3.1   Unified inpainting model

**Fusion parameterization**. Unlike previous methods, our method reconceptualizes the image fusion problem to enhance its robustness against registration errors. Precisely, as shown in Eq. 2, the distortion degree of $I_{wl}(x, y)$ is relatively low because it involves only minor warping based on $\mathtt{I}$.

This means that even in the presence of registration errors, $I_{wl}(x,y)$ does not introduce large-scale distortions. Therefore, we propose to construct a coarse fusion image $I_{CF}(x,y)$ via Eq. 4.

$$I_{CF}(x,y) = I_{wl}(x,y) + I_{wr}(x,y) \odot (1 - (M_{wl}(x,y) \& M_{wr}(x,y))), \tag{4}$$

where, $\&$ and $\odot$ denote the bitwise AND operators and element-wise multiplication operator. The coarse fusion image has noticeable seams, as shown in Figure 1(b). Also, with registration errors, incoherent image content appears around the seams. To solve this problem, we propose to focus on inpainting the image content around the seam, ensuring cohesion and coherence. Therefore, we use the seam mask $M_{seam}(x,y)$ to define the area in the fusion image that needs to be inpainted, as detailed in Eq. 5.

$$M_{seam}(x,y) = \texttt{Dilation}(M_{wl}(x,y), K_s) \oplus M_{wl}(x,y) \vee$$
$$\texttt{Erosion}((M_{wl}(x,y), K_s) \oplus M_{wl}(x,y) \& M_{wr}(x,y), \tag{5}$$

where, $\texttt{Dilation}(\cdot)$ and $\texttt{Erosion}(\cdot)$ denote the dilation and erosion operations [38], $K_s$ is the kernel sizes, $\vee$, $\oplus$ denote bitwise OR and XOR operators. Then, we inpaint $I_{CF}(x,y)$ based on seam mask $M_{seam}(x,y)$ and the inpainting function $f_\theta(\cdot)$ to obtain inpainted fusion image $\hat{I}_{CF}(x,y)$, as detailed in Eq. 6.

$$\hat{I}_{CF}(x,y) = I_{CF}(x,y) \odot (1 - M_{seam}(x,y)) + f_\theta(I_{CF}(x,y)) \odot M_{seam}(x,y). \tag{6}$$

**Rectangling parameterization**. Our method also defines the rectangling challenge as an inpainting problem based on the content mask $M_{content}(x,y)$. We use Eq. 7 to obtain the inpainted rectangling image $\hat{I}_{CR}(x,y)$.

$$\hat{I}_{CR}(x,y) = I_{CF}(x,y) \odot (1 - M_{content}(x,y)) + f_\theta(I_{CF}(x,y)) \odot M_{content}(x,y), \tag{7}$$

where, $M_{content}(x,y) = M_{wl}(x,y) \vee M_{wr}(x,y)$.

**Unified model**. Integrating Eq. 6 and Eq. 7, we obtain a unified inpainting model for fusion and rectangling, as shown in Eq. 8.

$$\hat{I}_{CFR}(x,y) = I_{CF}(x,y) \odot (1 - M_{inpaint}(x,y)) + f_\theta(I_{CF}(x,y)) \odot M_{inpaint}(x,y), \tag{8}$$

where, $M_{inpaint}(x,y) = M_{seam}(x,y) \vee M_{content}(x,y)$. By combining equations Eq. 3 and Eq. 8, this inpainting problem can be solved by a diffusion model, as detailed in Eq. 9.

$$\hat{\mathbf{x}}_{t-1} = \mathbf{x}_0 \odot (1 - M_{inpaint}(x,y)) + \mathbf{x}_{t-1} \odot M_{inpaint}(x,y), \tag{9}$$

where, $\mathbf{x}_0 = \mathcal{E}(I_{CF}(x,y))$, and $\mathbf{x}_{t-1} \sim \mathcal{N}(\mu_\theta(\mathbf{x}_t, t), \Sigma_\theta(\mathbf{x}_t, t))$.

### 3.2 Weighted mask guided reverse process

After defining the unified inpainting model for the fusion and rectangling tasks in the previous subsection, we discuss the method to control the inpainting strength in different regions during the reverse process, ensuring that both tasks can be accomplished in a single inference. Specifically, regions under the mask $M_{seam}(x,y)$ that contain the semantics of the original image are preserved as much as possible. In contrast, regions under $M_{content}(x,y)$ may require more powerful inpainting. We propose weighted masks to guide the reverse process to achieve this varying inpainting strength.

**Weighted masks**. Weighted masks are constructed from the weighted initial mask $\widetilde{M}_{init}(x,y)$ and inpainting mask $\widetilde{M}_{inpaint}(x,y)$.

---

**Algorithm 1** **W**eighted **M**ask **G**uided **R**everse Process (**WMGRP**)

---

1: **Input**: Coarse Fusion image $I_{CF}(x,y)$; Inference steps $N$; Radius $R$;
2:     Weighted initial mask $\widetilde{M}_{init}(x,y)$; Weighted inpainting mask $\widetilde{M}_{inpaint}(x,y)$
3:   prompt $p \leftarrow$ ""                              ▷ Our method does not require prompt guidance
4:   $I_{CFR}(x,y) \leftarrow \texttt{Telea}(I_{CF}(x,y), M_{content}(x,y), R)$                ▷ Coarse rectangling
5:   $\mathbf{x}_N \leftarrow \mathcal{E}(I_{CFR}(x,y))$                              ▷ Encode image
6:   // Based on the inpainting model, so there is a little difference here with the Eq. 9
7:   $\mathbf{x}_0 \leftarrow \mathcal{E}(I_{CFR}(x,y) \odot \widetilde{M}_{init}(x,y))$
8:   $\widetilde{M}_{inpaint}^{small}(x,y), \widetilde{M}_{init}^{small}(x,y) \leftarrow \texttt{DownSample}(\widetilde{M}_{inpaint}(x,y), \widetilde{M}_{init}(x,y))$
9:   $\mathbf{x}'_N \leftarrow \texttt{AddNoise}(\mathbf{x}_N, N)$
10:   $\hat{\mathbf{x}}_N \leftarrow \texttt{Concat}(\mathbf{x}'_N, \widetilde{M}_{init}^{small}(x,y), \mathbf{x}_0)$
11: **for** $t = N-1, \cdots, 0$ **do**                              ▷ Reverse process
12:   $\mathbf{x}'_t \leftarrow \texttt{DeNoise}(\hat{\mathbf{x}}_{t+1}, p, t)$
13:   $\widetilde{M}_t^{small}(x,y) \leftarrow 1 - (\widetilde{M}_{inpaint}^{small}(x,y) \preceq \frac{N-t}{N})$         ▷ $\preceq$ means element-wise less-than
14:   $\hat{\mathbf{x}}_t \leftarrow \texttt{Concat}(\mathbf{x}'_t, \widetilde{M}_t^{small}(x,y), \mathbf{x}_0)$
15: **end for**
16:   $\hat{I}_{CFR}(x,y) \leftarrow \texttt{ImageDecoder}(\hat{\mathbf{x}}_0)$                              ▷ Decode image
17: **Output**: $\hat{I}_{CFR}(x,y)$

---

The weighted initial mask $\widetilde{M}_{init}(x,y)$ assigns different fidelity levels to each pixel of the fusion image, determining how much to modify each pixel based on its fidelity during the reverse process. The formula of $\widetilde{M}_{init}(x,y)$ is given by Eq.10, which is composed of two parts. The left part determines the fidelity levels of pixels in $M_{seam}(x,y)$ region, and the right part determines the fidelity levels of pixels in $M_{content}(x,y)$ region.

$$\widetilde{M}_{init}(x,y) = \frac{\texttt{DT}(M_{seam}(x,y), K_g) \times \epsilon_1}{\max \texttt{DT}(M_{seam}(x,y), K_g)} \oplus \frac{\texttt{DT}(M_{content}(x,y), K_g) \times \epsilon_2}{\max \texttt{DT}(M_{content}(x,y), K_g)}, \qquad (10)$$

where, $\texttt{DT}(\cdot)$ is the distance transform operation [38] with kernel size $K_g$, $\epsilon_1$ and $\epsilon_2$ are hyperparameters.

The weighted inpainting mask $\widetilde{M}_{inpaint}(x,y)$, as described in Eq.11, is inspired by the suffix principle [24]. During the reverse process, $\widetilde{M}_{inpaint}(x,y)$ is mapped into multiple sub-masks to define the modified regions at each step $t$. The region corresponding to $M_{content}(x,y)$ contains no image content, so its size remains constant across all sub-masks, ensuring it is repainted throughout the entire process. Conversely, the region corresponding to $M_{seam}(x,y)$ contains a substantial quantity of original image information, and its size gradually increases with each step $t$, indicating that this region requires progressive modification. This gradual modification method facilitates a more seamless blending of the inpainting content with the original image content.

$$\widetilde{M}_{inpaint}(x,y) = M_{content}(x,y) \vee (1 - \texttt{DT}(M_{seam}(x,y), K_g)). \qquad (11)$$

**Guided reverse process**. We observe that when the missing region of the fusion image is large, the diffusion model very easily generates abnormal content, such as abnormal textures and words. To mitigate this issue, we introduce coarse rectangling. To be specific, we employ the Alexandru Telea Algorithm $\texttt{Telea}(\cdot)$ [46] to generate the coarse rectangling image: $I_{CFR}(x,y) = \texttt{Telea}(I_{CF}(x,y), M_{content}(x,y), R)$, where $R$ is the radius of a circular neighborhood of each point inpainted. The $\texttt{Telea}(\cdot)$ algorithm introduces a weak prior without any specific semantic information to the image $I_{CF}(x,y)$. As shown in the partial image $I$ of Figure 1, the image of the coarse rectangling region is completely blurred. The experimental result shows that generating images with weak priors significantly reduces the likelihood of producing anomalous

content compared to leaving blank areas entirely black. More details regarding the advantages of coarse rectangling can be found in the Appendix A.1.

The specific steps of the reverse process are detailed in Algorithm 1. Although this algorithm is based on the Stable Diffusion Inpainting model [41, 3], which differs slightly from the original Stable Diffusion model [2], the underlying principles remain consistent. In addition, our method works without the need for prompt, effectively reducing dataset requirements.

Appendix A.2 provide more detailed explanation and visualization of the weighted masks and WMGRP algorithm.

# 4 Experiments

## 4.1 Experimental setup

**Dataset**. To validate the performance of our method, we conducted experiments on the large public dataset UDIS-D [33]. To the best of our knowledge, UDIS-D is the only publicly available large-scale dataset in this field. Appendix D.4 provides more results of our method on other traditional small datasets.

**Baselines**. To our knowledge, no open-source solutions simultaneously address the fusion and rectangle stages of the image-stitching pipeline as comprehensively as our method. Table 1(a) gives brief statistics of related works, and more related work details provided in Appendix B. Therefore, we have to establish the comparison baselines by combining several existing methods. For the registration and fusion stage, we employ pre-trained models from UDIS [33] and UDIS++ [35]. For the rectangling stage, we utilize pre-trained models from DeepRectangling (DR) [34], Lama [44], Stable-Diffusion-v1-5-inpainting (SD1.5) [41], and Stable-Diffusion-v2-inpainting (SD2) [3]. Table 1(b) presents the detailed configurations of baselines.

Table 1: Statistics of related works and details of comparison baselines.

(a) Statistics of related works.

| Name | Stage1 | Stage2 | Stage3 |
|---|---|---|---|
| VFISNet [32] | ✓ | ✓ | ✗ |
| EPISNet [36] | ✓ | ✓ | ✗ |
| UDIS [33] | ✓ | ✓ | ✗ |
| UDIS++ [35] | ✓ | ✓ | ✗ |
| Dseam [11] | ✗ | ✓ | ✗ |
| Jiang et al. [22] | ✓ | ✓ | ✗ |
| LBHomo [21] | ✓ | ✗ | ✗ |
| RHWF [7] | ✓ | ✗ | ✗ |
| HomoGAN [18] | ✓ | ✗ | ✗ |
| DR [34] | ✗ | ✗ | ✓ |

(b) Details of comparison baselines.

| Baseline | Stage1 and 2 | Stage3 |
|---|---|---|
| UDIS+DR | UDIS | DR |
| UDISplus+DR | UDIS++ | DR |
| UDIS+Lama | UDIS | Lama |
| UDISplus+Lama | UDIS++ | Lama |
| UDIS+SD1.5 | UDIS | SD1.5 |
| UDISplus+SD1.5 | UDIS++ | SD1.5 |
| UDIS+SD2 | UDIS | SD2 |
| UDISplus+SD2 | UDIS++ | SD2 |

**Variants**. In this paper, we mainly present SRStitcher based on Stable Diffusion Inpainting model [3]. However, our method is versatile and can be readily adapted to other diffusion-based models with only minor modifications. In the experiments, we also compare the SRStitcher variants, including: SRStitcher-S based on the Stable Diffusion 2 model [2], SRStitcher-U based on Stable Diffusion 2 Unclip model [4], SRStitcher-C based on Controlnet Inpainting model [54]. For further information on SRStitcher variants, please refer to Appendix D.3.

**Metrics**. (1) **Stitched image quality**. Since UDIS-D is an unsupervised dataset, we use the No-Reference Image Quality Assessment (NR-IQA) metrics to evaluate the image quality. Specifically, we use the HIQA [43] and CLIPIQA [49]. (2) **Content consistency**. We develop a new metric to evaluate the content consistency between the input images and the stitched image. Specifically, we introduce the $\texttt{CoCa}(\cdot)$ [50] model and $\texttt{Bert}(\cdot)$ [39] model to extract text from the images and generate text embeddings. The similarity between these embeddings is measured by the cosine similarity $\texttt{cosine}(\cdot)$. We design the Content Consistency Score (CCS) metric:

$$CCS = (CCS_n + CCS_g)/2 \tag{12}$$

$CCS_n$ measures the local consistency, which compares the stitched image $I_{Stitched}(x, y)$ and the fusion image $I_{Fusion}(x, y)$. Both images are split into $n$ equal parts for detailed comparison: $CCS_n = \texttt{cosine}(\Upsilon(\texttt{Split}(I_{Stitched}(x, y), n)), \Upsilon(\texttt{Split}(I_{Fusion}(x, y), n)))$, where $\Upsilon(\cdot) = \texttt{Bert}(\texttt{CoCa}(\cdot))$, and for this test, $n = 4$. In addition, $CCS_g$ assesses the overall content consistency between the $I_{Stitched}(x, y)$ and original input images: $CCS_g = \texttt{cosine}(\Upsilon(I_{Stitched}(x, y), \Upsilon(I_l(x, y), I_r(x, y)))$. Please refer to Appendix C for further information on the metrics.

**Implement details**. All experiments are performed on a single NVIDIA 4090 GPU. In addition, all experiments of SRStitcher described in this paper are based on these pre-aligned images made by UDIS++ [35]. For hyper-parameters, the guidance scale and inference steps $N$ are set to 7.5 and 50; The $K_s$ in Eq. 5 is set to $\lceil W^*/\lambda \rceil \times \delta$, where $\lambda = 200$ and $\delta = 10$; The $K_g$ in Eq. 10 and Eq. 11 are set to 3; The $R$ in $\texttt{Telea}(\cdot)$ is set to 20. The $\epsilon_1$ and $\epsilon_2$ in Eq. 10 are set to 128 and 128.

Table 2: Quantitative results. The best and second-best results are highlighted by red and blue. $\star$ refers to the inference results of this method are not affected by seed. $\dagger$ means the inference results of this method are affected by the seed. We tested the results five times by varying the seed, taking the average and standard deviation.

| Method | $UDIS - D_{test}$ | | | $UDIS - D_{train}$ | | |
|---|---|---|---|---|---|---|
| | HIQA ↑ | CLIPIQA ↑ | CCS(%)↑ | HIQA ↑ | CLIPIQA ↑ | CCS(%) ↑ |
| UDIS+DR⋆ | 42.53 | 28.33 | 89.35 | 45.31 | 31.29 | 90.02 |
| UDISplus+DR⋆ | 45.98 | 31.24 | 88.45 | 49.87 | 33.47 | 90.69 |
| UDIS+Lama⋆ | 42.55 | 27.17 | 84.99 | 45.63 | 30.15 | 86.70 |
| UDISplus+Lama⋆ | 46.57 | 31.48 | 87.73 | 51.28 | 33.29 | 86.12 |
| UDIS+SD1.5† | 42.60 | 28.03 | 87.42 | 48.59 | 28.57 | 87.74 |
| | ± 2.24 | ± 2.84 | ± 1.08 | ± 1.18 | ± 0.89 | ± 1.36 |
| UDISplus+SD1.5† | 46.45 | 27.13 | 87.16 | 50.89 | 30.16 | 88.12 |
| | ± 1.11 | ± 1.85 | ± 1.61 | ± 2.20 | ± 1.46 | ± 1.35 |
| UDIS+SD2† | 42.84 | 28.00 | 85.97 | 47.15 | 34.31 | 85.72 |
| | ± 1.05 | ± 0.89 | ± 1.33 | ± 1.33 | ± 0.95 | ± 1.55 |
| UDISplus+SD2† | 46.98 | 31.23 | 89.37 | 51.49 | 34.26 | 91.18 |
| | ± 1.43 | ± 2.18 | ± 1.23 | ± 1.74 | ± 1.24 | ± 1.35 |
| *SRStitcher Variants* | | | | | | |
| SRStitcher-S† | 45.66 | 32.08 | 85.91 | 51.73 | 35.23 | 87.32 |
| | ± 0.89 | ± 0.91 | ± 0.74 | ± 0.56 | ± 0.79 | ± 0.81 |
| SRStitcher-U† | 43.89 | 28.35 | 85.81 | 48.18 | 31.38 | 86.33 |
| | ± 1.01 | ± 0.66 | ± 1.01 | ± 0.55 | ± 0.74 | ± 0.53 |
| SRStitcher-C† | 46.57 | 31.34 | 89.47 | 52.73 | 34.53 | 91.41 |
| | ± 0.89 | ± 0.76 | ± 0.71 | ± 0.74 | ± 0.85 | ± 0.84 |
| SRStitcher† | **47.82** | **33.25** | **91.15** | **54.74** | **37.52** | **93.29** |
| | ± 0.55 | ± 0.57 | ± 0.52 | ± 0.63 | ± 0.68 | ± 0.45 |

## 4.2 Quantitative evaluation

We perform a comprehensive quantitative analysis by comparing the results of 10,440 sample pairs from the UDIS-D training set $UDIS - D_{train}$ and 1,106 sample pairs from the testing set $UDIS - D_{test}$. Notably, our method does not require training, so to provide a broader base of comparison, the training set of UDIS-D is also included in the comparison experiments. The comparative results are presented in Table 2, which illustrates the significant advantages of SRStitcher in terms of the stitched image quality and content consistency.

## 4.3 Qualitative evaluation

We perform a quantitative evaluation of SRStitcher against other baseline methods, depicted in Figure 2. The first row of Figure 2 presents a challenging registration scenario involving soft and deformable objects, such as wires, which may have deformed unpredictably between two images. Current registration methods cannot accurately align such objects. Instead of attempting to register or fuse these deformed wires, our method opts to *inpaint incorrect wires*, effectively overcoming registration errors. A more detailed discussion on this is available in Appendix A.4. The second row of Figure 2 illustrates challenges associated with structured and extensive missing areas, where methods like DR and Lama struggle to accurately fill in the image content. The third row addresses the repeated pattern challenge, where a large number of bricks significantly complicates registration accuracy. The fourth row highlights the classic multi-depth layer problem, illustrating how objects like pillars and their backgrounds, being on different depth layers, result in registration inaccuracies. To enhance the clarity of presentation, the results of UDISplus+DR, UDIS+Lama, UDIS+SD1.5, and UDISplus+SD1.5 are omitted from this figure. Detailed qualitative evaluations for all comparison methods are provided in Appendix D.

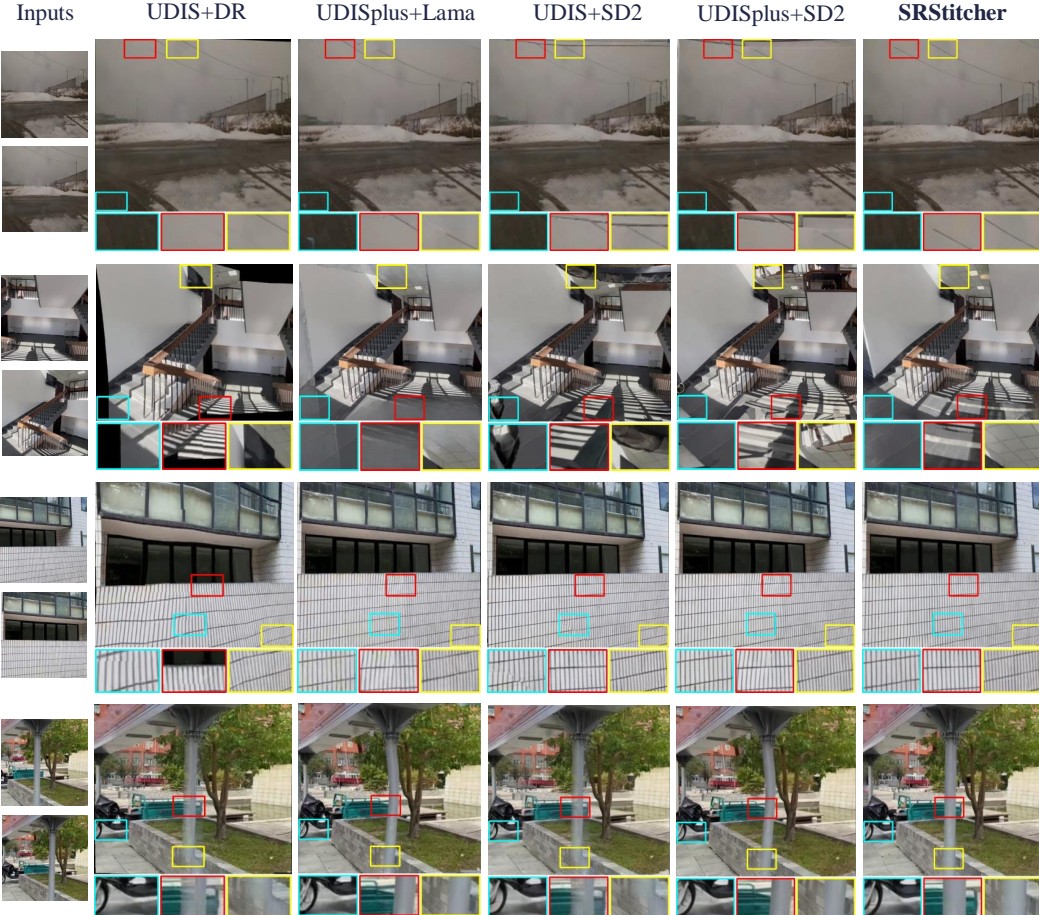

Figure 2: Qualitative evaluation results. All visual results are obtained with seed 0.

## 4.4 User study

We introduce a user study metric from UDIS [33]. This method allows for a more subjective but insightful visual quality assessment through direct user feedback. For the user study, we display four images simultaneously on a single screen: the two input images, our stitched result, and the stitched result from one of the baseline methods. Participants are asked to determine which result is superior, *SRStitcher* or *Another* (comparison baseline). If a clear preference is not apparent, participants

can choose *Both Good* or *Both Bad*. The study involves 20 participants: 10 researchers (computer vision background) and 10 volunteers (non-computer major). This diverse group ensures a balanced perspective, combining expert technical evaluation with general user impressions. The results are shown in Figure 3.

## 4.5 Ablation study

**Hyper-parameter**. Figure 4 illustrates the ablation results in hyper-parameter of SRStitcher, demonstrating that these parameters are highly interpretable and easy to adjust. (1) $\lambda$**: Controls the width of the region in** $M_{seam}$. A smaller $\lambda$ increases the modification range and decreases the CCS. For stitched pictures with color differences, a lower $\lambda$ value can better fuse the images. We set $\lambda$ to 200, considering both image smoothness and CCS. (2) $R$**: Controls the granularity of the coarse rectangling image**. A higher $R$ value provides a higher quality weak prior for inpainting, reducing the likelihood of generating abnormal content. Ideally, a larger $R$ is preferable, but due to the limitations in GPU acceleration with the $\texttt{Telea}(\cdot)$, a very high $R$ value can slow down the pipeline. Thus, we balance performance and speed by setting $R$ to 20. (3) $\epsilon_1$**: Controls the inpainting strength of the seam area**. At $\epsilon_1 = 128$, the shape of the pillars appears more reasonable compared to $\epsilon_1 = 64$. However, increasing $\epsilon_1$ to 192 significantly alters the image content, so we set it to 128. (4) $\epsilon_2$**: Controls the inpainting strength of the rectangling area**. When $\epsilon_2$ is relatively low, the image structure remains largely intact, but increasing it to 192 leads to noticeable structural deficits. Therefore, we set $\epsilon_2$ to 128. Please see Appendix D.2 for more comprehensive hyper-parameters studies.

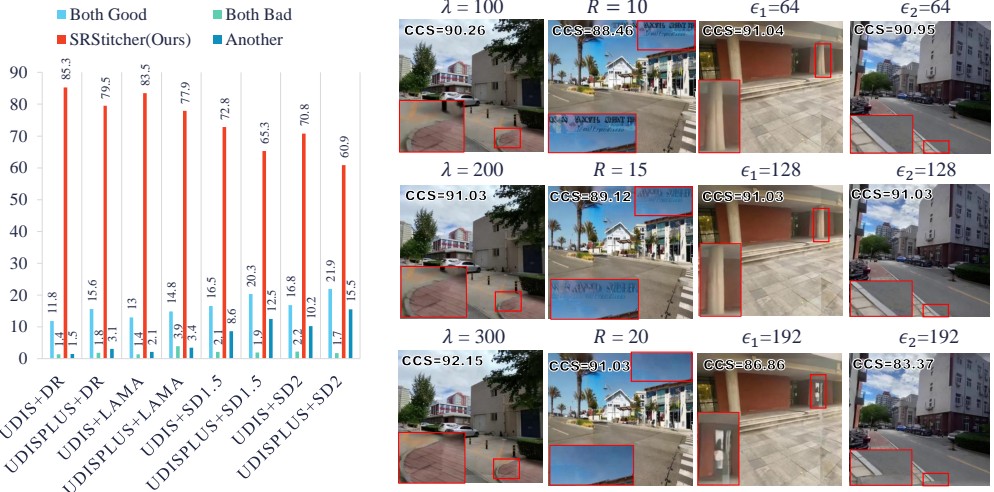

Figure 3: User study on visual quality. The results are averaged across 20 participants, with the percentage on the ordinate axis.

Figure 4: Ablation study results. CCS on each image is the average score of $UDIS - D_{test}$ with this hyper-parameter and seed 0, not a single image.

**Weighted Masks**. Figure 5 illustrates the ablation results in different masks guided manners. Specifically, Figure 5(a) represents the different effects of using fixed mask and weighted mask in the fusion region. As shown in the red box, the weighted mask better smoothes the image content while preserving the original information of the image by gradually modifying the image content, while the fixed mask significantly modifies the original image content. In addition, Figure 5(b) illustrates the stitched images obtained by the gradually changing weighted masks in the rectangling region. Since the rectangling region contains no image content, the generator guided by the gradually changing masks repeatedly smoothes the empty region, resulting in blurred noise. Therefore, we use a fixed size mask in the rectangling region.

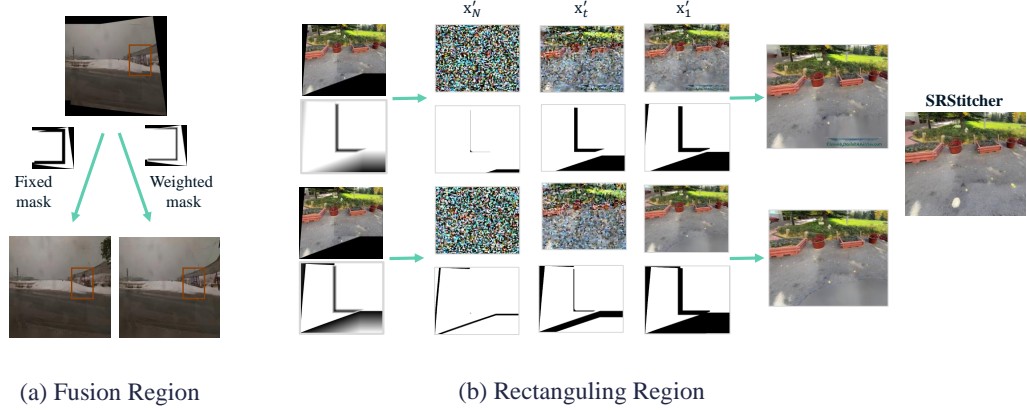

(a) Fusion Region    (b) Rectanguling Region

Figure 5: Ablation of the masks guided manners.

## 5 Discussion and conclusion

This paper introduces SRStitcher, which reconceptualizes the fusion and rectangling stages as a unified inpainting model. Through weighted masks, SRStitcher leverages the robust generalization capabilities of a pre-trained large-scale generation model to accomplish this complex inpainting task without additional fine-tuning or task-specific data annotations. Extensive experiments demonstrate that SRStitcher significantly outperforms existing state-of-the-art methods regarding the quality of the stitched images and its robustness to registration errors and abnormal content. Furthermore, the specific effects and adjustments of each hyper-parameter in SRStitcher are detailed in the ablation studies, illustrating its high interpretability and controllability.

However, there are still some limitations and open issues in future research: (1)**Visible seam**. When input images exhibit significant color differences, visible seams may appear with the parameter settings described in the paper. Adjustments to $\epsilon_1$ and $\lambda$ can partially mitigate this issue, but such modifications can compromise the preservation of original image information. We speculate that a more flexible and appropriately designed hyper-parameter selection scheme could solve this problem. (2) **Local blurring**. We use coarse rectangling and $\widetilde{M}_{init}(x, y)$ to control the content generation. However, this approach introduces a side effect where some challenging scenes appear locally blurred (See Appendix D.5). This issue presents a dilemma: accept local blurring or risk producing anomalous images. We temporarily choose to tolerate local blurring. Future improvements will include a refined coarse rectangling approach or fine-tuning the model. (3) **Integrating registration**. Is it possible to integrate the registration stage into the unified model? According to Diffusion Features (DIFT) [45], it is possible. DIFT proves that the geometric correspondence between images can be effectively established by extracting feature maps from their intermediate layers at a specific timestep during the inverse process. Replacing the registration method used by SRStitcher with DIFT is straightforward. However, our ambitions extend beyond simple replacement. We believe there is potential for a more elegant and concise method to integrate concepts proposed by DIFT into our existing method.

## 6 Acknowledgment

This paper was supported by the National Key Research and Development Project of China (Grant No.2023YFB3408600) and Science and Technology Innovation Program of Shanghai (Grant No.18DZ2295100).

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

# Appendix

## A More details of SRStitcher

### A.1 More analysis of the designs described in the main paper

To elucidate the specific role of each design element in SRStitcher, Figure 6 illustrates the results after sequentially removing our designs:

(a) **SRStitcher result.** The result is obtained when coarse rectangling, $\widetilde{M}_{init}(x, y)$, and $\widetilde{M}_{inpaint}(x, y)$ are all used.

(b) **Remove coarse rectangling, retain $\widetilde{M}_{init}(x, y)$ and $\widetilde{M}_{inpaint}(x, y)$.** Removing coarse rectangling while maintaining $\widetilde{M}_{init}(x, y)$ results in the incomplete filling. As mentioned above, this is because $\widetilde{M}_{init}(x, y)$is still working and retains the original image information. But, the area previously filled by the coarse rectangling returns pure black, which affects the final stitched result.

(c) **Remove coarse rectangling and $\widetilde{M}_{init}(x, y)$, retain $\widetilde{M}_{inpaint}(x, y)$.** Replacing $\widetilde{M}_{init}(x, y)$ with $M_{content}(x, y)$ eliminates its effect, allowing the rectangling area to be completely filled. However, compared to (a), the content changes significantly, and abnormal content emerges. This result indicates the importance of coarse rectangling in providing weak priors for guiding the generation.

(d) **Remove coarse rectangling, $\widetilde{M}_{init}(x, y)$, and $\widetilde{M}_{inpaint}(x, y)$.** By removing all design elements, SRStitcher becomes a simple inpainting model based on $M_{inpaint}(x, y)$. This results in a higher probability of generating abnormal content, with substantial alterations near the seam (indicated by the red box).

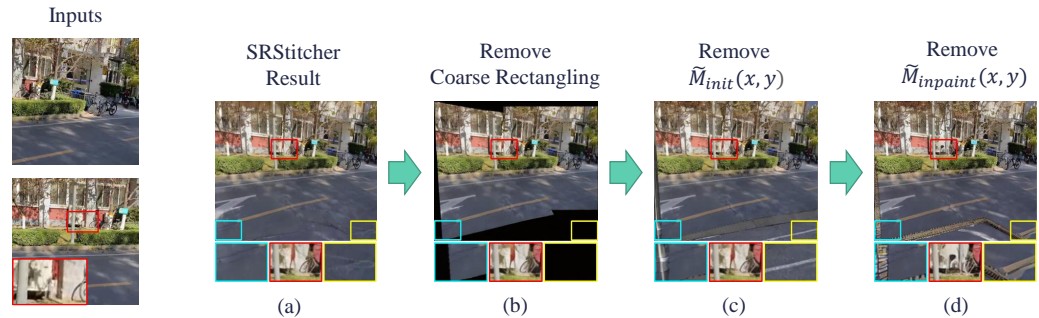

Figure 6: The effects of each design on SRStitcher results. We demonstrate how each design element influences the results of SRStitcher by removing them one at a time and observing the changes.

### A.2 Physical implications of weighted masks $\widetilde{M}_{init}(x, y)$ and $\widetilde{M}_{inpaint}(x, y)$

We design $\widetilde{M}_{init}(x, y)$ and $\widetilde{M}_{inpaint}(x, y)$ is illuminated by the following observations:

(1) **The input structure of the inpainting model**. Unlike the general Stable Diffusion model[2], the input of the Stable Diffusion Inpainting model comprises the original image, mask and masked image. Through experimental tests, we found that variations in any part of this composite input significantly influence the final output results.

(2) **Impact of masked image**. The masked image retains and continuously provides the unmasked area information of the original image throughout the reverse process, ensuring that the unmasked area of the final generated image remains consistent with the original image.

(3) **Impact of mask**. The mask is employed to identify the modified regions during the reverse process. Adjusting the scope of the mask during the process allows for the controlled modification of the inpainting intensity across different regions. One point to note is that in the input mask of the

Stable Diffusion Inpainting model: *the black is the area that needs to be retained, and the white is the area that needs to be inpainted.*

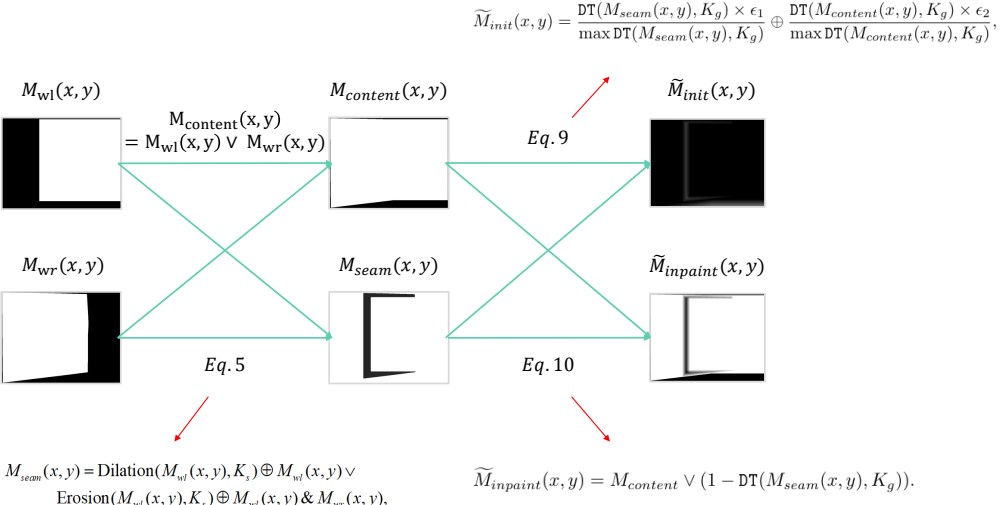

$$\widetilde{M}_{init}(x,y) = \frac{\mathtt{DT}(M_{seam}(x,y),K_g) \times \epsilon_1}{\max \mathtt{DT}(M_{seam}(x,y),K_g)} \oplus \frac{\mathtt{DT}(M_{content}(x,y),K_g) \times \epsilon_2}{\max \mathtt{DT}(M_{content}(x,y),K_g)},$$

$M_{wl}(x,y)$

$M_{content}(x,y)$
$= M_{wl}(x,y) \vee M_{wr}(x,y)$

$M_{content}(x,y)$

$Eq.9$

$\widetilde{M}_{init}(x,y)$

$M_{wr}(x,y)$

$M_{seam}(x,y)$

$Eq.5$

$Eq.10$

$\widetilde{M}_{inpaint}(x,y)$

$$M_{seam}(x,y) = \mathrm{Dilation}(M_{wl}(x,y),K_s) \oplus M_{wl}(x,y) \vee \\ \mathrm{Erosion}(M_{wl}(x,y),K_s) \oplus M_{wl}(x,y) \& M_{wr}(x,y),$$

$$\widetilde{M}_{inpaint}(x,y) = M_{content} \vee (1 - \mathtt{DT}(M_{seam}(x,y),K_g)).$$

Figure 7: Visual production of masks. We provide the correlation between equations and masks to facilitate comprehension.

Based on the above findings, we propose the construction of weighed masks.

(1) **Control of masked image**. In our design, $\widetilde{M}_{init}(x,y)$ is used to create the masked image, determining the extent to which information from the original image is retained. We aim to fully preserve the image content in $M_{content}(x,y)$. While, for areas outside $M_{content}(x,y)$(that is the coarse rectangling area), we implement a distance transform to change the $M_{content}(x,y)$. Due to the coarse rectangling regions are blurry (but we must use the weak prior, as explained in A.1), it is undesirable to retain substantial information from these blurry regions. By employing a distance transform, we can *gradually reduce the information* from the coarse rectangling image starting from the edges of $M_{content}(x,y)$. This approach ensures that only the most relevant information from the edges of the optimal coarse rectangling is retained, avoiding the use of excessive coarse rectangling data.

(2) **Control of mask**. In our design, the inpainting mask is dynamically adjusted throughout the reverse process based on $\widetilde{M}_{inpaint}(x,y)$. Although $\widetilde{M}_{inpaint}(x,y)$ serves as just one mask, its mask area is modified at each step $t$ by calculating the threshold $\frac{N-t}{N}$ and remapping $\widetilde{M}_{inpaint}(x,y)$ to $\widetilde{M}_t^{small}(x,y)$ accordingly (as detailed in Algorithm 1).

Figure 7 provides the detailed mask productions. The final $\widetilde{M}_{init}(x,y)$ and $\widetilde{M}_{inpaint}(x,y)$ contain gradient areas, and we realize the gradual control of the reverse process based on the gradient areas. The $\widetilde{M}_{init}(x,y)$ is used to store information about the fusion image. Intuitively, darker places hold more information about the original image.

### A.3 Visualization of WMGRP

As a supplement to Algorithm 1, we provide Figure 8 to visually illustrate the process of WMGRP using the masks generated from Figure 7. The $\widetilde{M}_{inpaint}(x,y)$ is employed to regulate the intensity of inpainting in distinct regions during the reverse process. Intuitively, the white part is *not modified at all*, and the black part is *modifiable*. It can be observed that the black area of inpainting mask is gradually increased during the reverse process, which serves as the guiding process for the gradual modification.

Although the above content is mainly based on the observations and experimental results of the Stable Diffusion Inpainting model, our experiments have proved that it also applies to the general

Stable Diffusion model [2] with only minor modifications. We provide the detailed implementation of SRStitcher variants in the Supplementary Material.

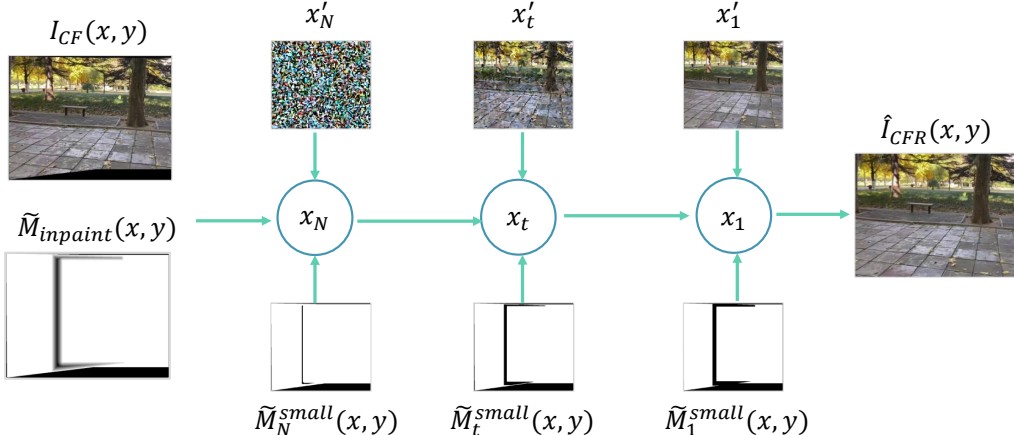

Figure 8: Visualization of WMGRP. For simplicity, we omit the masked image, which is invariant after initialization throughout the process.

## A.4 Why is SRStitcher so effective at overcoming registration errors

Unlike the previous fusion and rectangling methods SRStitcher does not rigidly adhere to the registration results. Figure 9 provides a clear example of how SRStitcher addresses wire registration errors. In the coarse fusion image $I_{CFR}(x, y)$, the misregistration problem is still serious, which is reflected in the significantly misaligned wires.

After the inpainting, these misaligned wires are effectively corrected, while the content in other masked areas remains largely unchanged. We attribute this remarkable correction capability to the strong generalization ability of large-scale generative models. This ability to correct incorrect image content underpins our motivation for employing the large-scale pre-trained model.

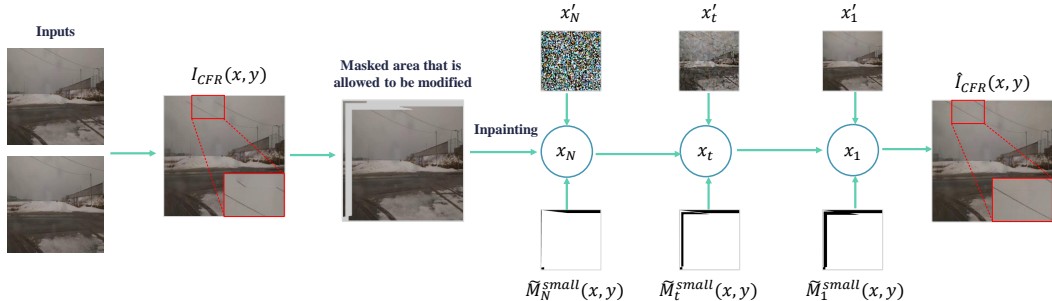

Figure 9: How SRStitcher addresses the issue of registration errors. Due to the disparate parameter settings, the figure differs from the main paper. The display effect and parameters of the main paper shall prevail.

For visual illustration, Figure 10 compares the fusion effects of SRStitcher and other methods in a large-parallax scene. The result shows that, SRStitcher's inpainting-based fusion method achieves the best continuity of image content in this challenging scenario.

## A.5 The necessity of using a large-scale diffusion model

Although the diffusion model has been shown to outperform Generative Adversarial Network (GAN) methods on the inpainting problem [30], its high hardware requirements have prevented us from

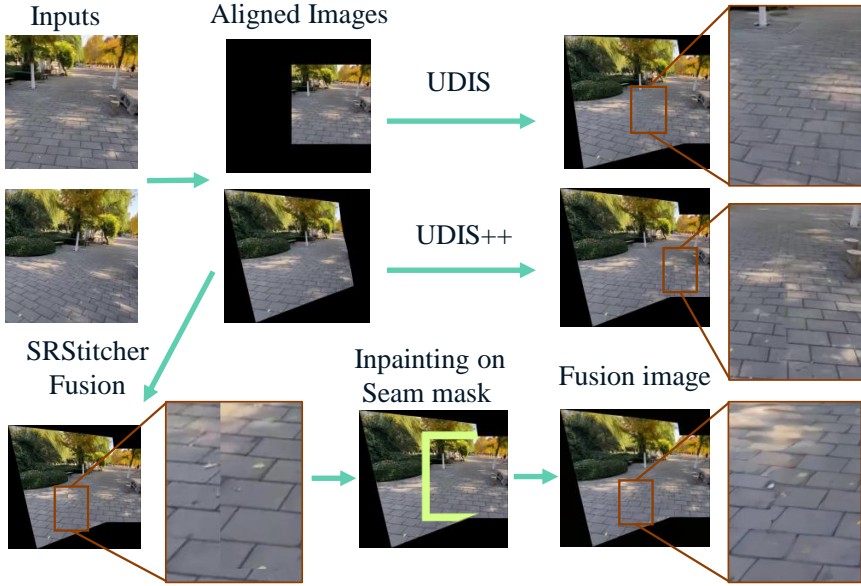

Figure 10: Comparison of fusion effects of SRStitcher and other methods in the large-parallax scene.

considering it as a first option. Initially, we try to address the inpainting problem using Generative Adversarial Network (GAN)-based methods, such as AOTGAN [53], Lama [44], and FCFGAN [20].

However, our experiments revealed several shortcomings. GAN-based models struggled with poor generalization ability, displayed unstable training outcomes, and demanded high-quality, well-labeled datasets. These factors complicated the pipeline and increased the workload, particularly in data labeling.

During the research bottleneck period, we found the concept of Soft-Inpainting proposed by Differential Diffusion [24], which involves blending parts of the image with the original by making slight modifications. This concept inspired us to adapt and extend it to our needs. We applied *Soft-Inpainting* near the seams and employed more intense *Hard-Inpainting* in the rectangling areas. With this idea, we successfully implemented SRStitcher proposed in this paper.

Moreover, as mentioned in the previous section, the strong generalization ability of large-scale diffusion models is also vital to implementing our method, which is the key to the ease of implementation of SRStitcher without training or task-specific data annotations.

Thus, adopting the diffusion model proved essential for addressing the challenges of this paper. It provided a more fitting solution than any other model available.

### A.6    Why not focus on the registration stage

The registration stage is not the primary focus of this paper. We employ a simplified homography estimation network from UDIS++ [35] to address the registration challenges. It is essential to clarify that this does not imply a devaluation of the registration stage. Registration has been the most extensively researched of the three stitching stages, with significant work devoted to improving homography accuracy. However, perfect homography matrices that precisely align images do not exist for scenes that are non-planar or involve cameras with different projection centers.

There are two mainstream methods to overcome these inherent limitations: the multi-homography warp method [52] and the dense match method [47]. However, the multi-homography method faces challenges in parallelization and integration within deep learning frameworks [35], while dense matching is generally slower and less robust.

These limitations inform our decision to leverage the existing homography network and concentrate our efforts on enhancing the robustness of the subsequent stages. Our experimental results show that

this decision has been beneficial: SRStitcher exhibits greater robustness to registration errors, thereby reducing the precision requirements of the registration stage.

## B   A brief survey of the image stitching pipeline

The image stitching pipeline can be divided into three stages, and the following subsections are described based on each stage.

### B.1   Image registration

Early image registration works [57, 8, 31] are limited by the feature extraction method, which often falters under conditions of rotation, scaling, and illumination changes. To solve the scale changes problem, AutoStitch [6] marks a significant advancement by incorporating the Scale-invariant Feature Transform (SIFT) to extract scale-invariant features. However, this method is challenging to apply to situations with multiple depth layers. To address multi-depth layers condition, DHW [12] proposes a model that assumes the presence of two distinct planes within the image, applying different homography adjustments to each. However, the performance of this method can be severely impacted by the dynamics of camera movement. More recently, NIS [28] introduces the depth map integration to enhance registration accuracy. However, this method relies on accurately estimating depth maps, presenting its own implementation challenges. Yu et al. [51] develop a technique using the epipolar displacement field to improve registration in scenes with significant parallax.

Feature-based methods have traditionally been the cornerstone of image registration techniques. However, these methods often need more geometric structure and in low-texture scenarios where traditional feature detection techniques are prone to failure.

In recent years, the advent of deep learning has revolutionized the field of image registration by enabling the extraction of rich semantic features through deep neural networks. Hoang et al [17] and Shi et al. [42] both propose the use of Convolutional Neural Networks (CNNs) to enhance feature representations in image stitching registration. Despite their progress, these approaches primarily use deep learning for feature enhancement rather than creating a holistic learning-based framework. VFISNet [32] is the first complete learning-based framework for image stitching, but it is limited by its inability to handle images of arbitrary resolutions. EPISNet [36] is improved on VFISNet by introducing a flexible mechanism that supports the input of any image size through scalable image and homography adjustments. HomoGAN [18] introduces a method based on the Generative Adversarial Network(GAN) to enhance the quality of homography estimations, representing a novel application of GANs in this field. Jiang et al. [22] integrates graph convolutional networks into the image stitching framework to boost the precision of multi-spectral image registration. LBHomo [21] introduces a semi-supervised approach to estimate homography more accurately in large-baseline scenes by sequentially multiplying multiple intermediate homography. RHWF [7] introduces homography-guided image warping and Focus transformer into the recursive homography estimation framework to further refine homography estimation accuracy.

### B.2   Image fusion

The earliest fusion method is weighted fusion [5], which requires high registration accuracy. If registration is imperfect or there is a color mismatch between the images, visible seams may appear, which can degrade image quality. APAP [52] introduces a smoothly varying projection field to enhance fusion accuracy. However, APAP tends to introduce severe perspective distortions in non-overlapping areas, limiting its applicability. Inspired by interactive digital photomontage [1], Gao et al. [13] propose the seam-based fusion method, which involves a seam prediction stage to identify optimal seam lines between overlapping images. Although effective, it is notably time-consuming. Therefore, SEAGULL [29] proposes to improve the previous seam-based methods by using estimated seams to guide local alignment optimization, enhancing seam quality and reducing processing time. However, it struggles with repetitive textures, where it still shows poor performance.

The methods above are all traditional fusion methods characterized by limited versatility and difficulty adapting to complex scenarios. To solve the defects of traditional solutions, UDIS [33] proposes a reconstruction-based model to improve the quality of the fused image. This method sometimes produces artifacts and strange blurs in overlapping areas despite its advances. Inspiration from

traditional seam-based approaches, UDS++ [35] and Dseam [11] both use deep learning to refine the seam finding process. Though these fusion methods offer more robust and flexible solutions to improve the ability to handle complex scenarios, they cannot still correct the registration error effectively.

## B.3 Image rectangling

Image rectangling is a relatively new area of computer vision with limited research to date. Prior to the advent of deep learning in this domain, traditional solutions such as those proposed by He et al.[15] and Li et al. [25] used mesh-based warping techniques to address missing areas in images. DeepRectangling [34] represents the first deep learning-based approach in image rectangling, accompanied by a baseline and a public dataset tailored for this specific task. The method continues to rely on mesh-based warping but incorporates learning algorithms to enhance the fill quality and handle complex scenarios more effectively. While these methods are groundbreaking, they often change the global relative pixel positions, which could lead to suboptimal results, especially in cases with large missing areas, resulting in incomplete fills. A more recent method RecDiffusion [56] that employs a diffusion model to better solve the rectangling. Although this method provides a sophisticated solution for achieving rectangularity, it is complex in design, requires long inference times, and requires significant computational resources for training, which limits its practical applicability.

Moreover, the current deep learning-based image rectangling methods are all based on the DIR-D dataset [34]. DIR-D dataset is a strong assumption dataset, which assumes that some challenging scenes are excluded and that the image registration and fusion are flawless. Therefore, current method do not optimize for the robustness of registration and fusion errors, leading to the error propagation problem illustrated in Figure 1 ③.

In addition, there are some works on filling irregular edges in other fields [40, 27, 23]. However, due to significant differences between these fields and the varying shapes of missing regions, these studies are not easily applicable to the rectangling task in the current image stitching pipeline.

## B.4 Other similar work

Several prior studies have attempted to integrate fusion and rectangling stages. For instance, RDISNet [55] claims to create an end-to-end image stitching framework that combines all three stages. However, our analysis of its network structure and experimental results suggests that RDISNet may struggle with large parallax scenes, and noticeable noise in its stitching outputs adversely impacts image quality. Chen et al. [10] propose a diffusion-based method to address both fusion and rectangling tasks. However, this method necessitates meticulously prepared datasets and retraining of the diffusion model, leading to substantial cost demands. In summary, while these existing methods are innovative, they present significant limitations with low reproducibility. As a result, we exclude them from our baseline comparisons.

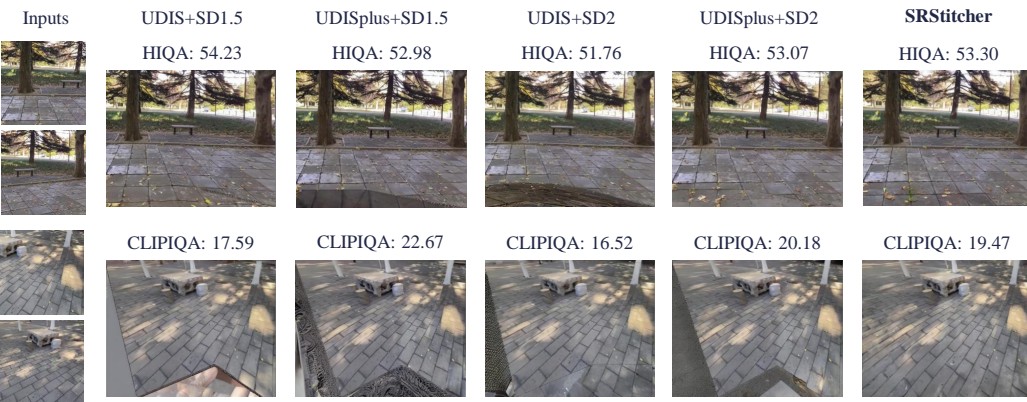

Figure 11: Examples of unreasonable NR-IQA scores.

# C Detail of metrics

## C.1 NR-IQA metric

### (1) NR-IQA metric settings

**HIQA**. HIQA [43] is designed for the 'wild' image. HIQA is particularly suitable for evaluating the predominantly outdoor images in the UDIS-D dataset, making it an ideal choice for our analysis. We implement this metric based on the public source code with default parameters.

**CLIPIQA**. CLIPIQA [49] based on the Contrastive Language-Image Pre-training (CLIP) models, which allows for adaptable evaluations across different datasets. We use IQA-PyTorch Tool [9] to implement this metric with prompts ['nature image', 'stitched image'] to evaluate whether the stitched images appear more natural.

### (2) Limitation of NR-IQA metrics

Through our analysis of the results obtained by the NR-IQA metrics HIQA and CLIPIQA, we find a discrepancy between these metrics and human sensory preferences for image quality. Sometimes, our method produces visually higher quality and authenticity images, as shown in Figure 11, but the scores assigned by these NR-IQA metrics are counterintuitively low.

We believe that this problem arises from a mismatch between the training datasets and the unique challenges of image stitching. Metrics such as HIQA and CLIPIQA, are trained on IQA-specific datasets such as KonIQ-10k [19] and Live-iWT [14], which focus primarily on image distortions such as white noise, low-light noise, and JPEG compression artifacts. These types of distortions are very different from those encountered in image stitching, such as artifacts and incongruous inpainting content. As a result, models trained on such data may struggle to perfectly reflect the true perceptual quality of stitched images.

While the NR-IQA metric may exhibit inaccuracies in certain scenarios, it typically indicates superior stitching images in most cases, which is why we have chosen to use it. However, we acknowledge the inherent limitations of current NR-IQA metrics, especially their inability to effectively capture the nuanced aspects of image quality improvements in stitching. Therefore, we do not employ these metrics in the ablation experiments. And, we also conduct user study as a supplementary proof.

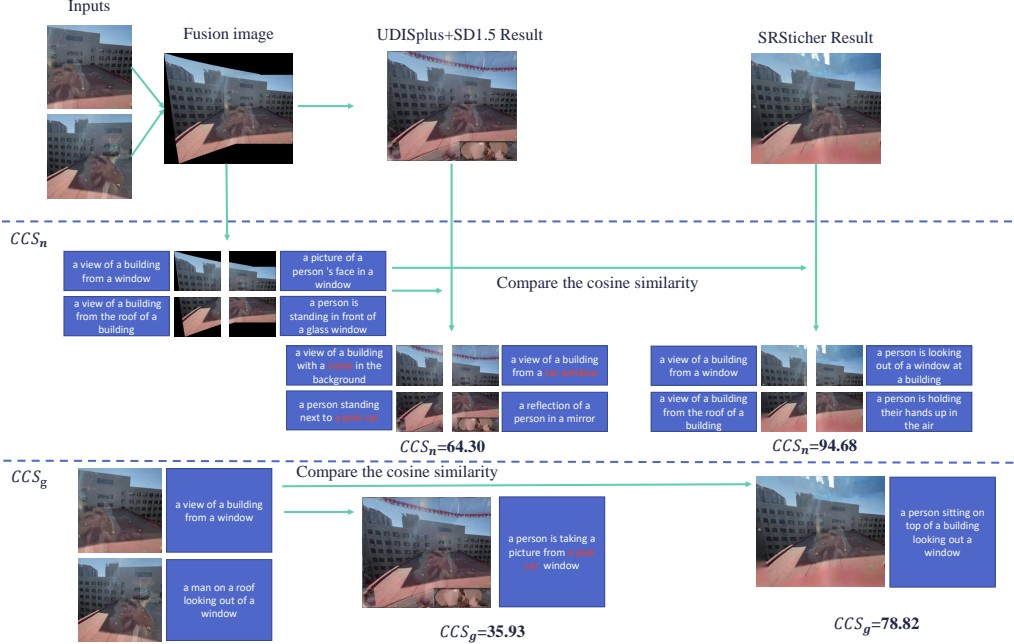

Figure 12: Visual presentation of the CCS metric. Inside the blue box is the text extracted using CoCa($\cdot$).

### C.2 CCS metric

We design the CCS metric to measure the image content consistency before and after stitching, which is based on the idea of an image-to-text model. We first extract text information based on the image through the $\texttt{CoCa}(\cdot)$ model, then map the text into the embedding space through the $\texttt{Bert}(\cdot)$ model, and finally measure the cosine similarity of the embedding before and after stitching to calculate the CCS. We give an intuitive evaluation of this metric in Figure 12.

In defining $CCS_n$, we set $n$=4, a value we believe is most effective. Using $n$=1 loses local meaning. For $n \geq 9$, in scenes with large areas missing (as illustrated in the second scene of Figure 2), parts of the local image may lack semantic content. This absence hampers the extraction of text information, thus undermining the credibility of the CCS metric. Additionally, excessively small input images can negatively impact the performance of the $\texttt{CoCa}(\cdot)$ model.

We also give the source code for the CCS metric implementation in the Supplementary Material, as shown in *metrics/ccs.py*.

## D  Additional experiments and results

### D.1  Evaluation results of the example in Figure 2 on all baselines

To ensure the presentation effect, all baseline results are not provided in the main paper Figure 2. Here, we offer the complete results, as shown in Figure 13.

Our results in the figure can also be reproduced using the source code and provided data in the Supplementary Material.

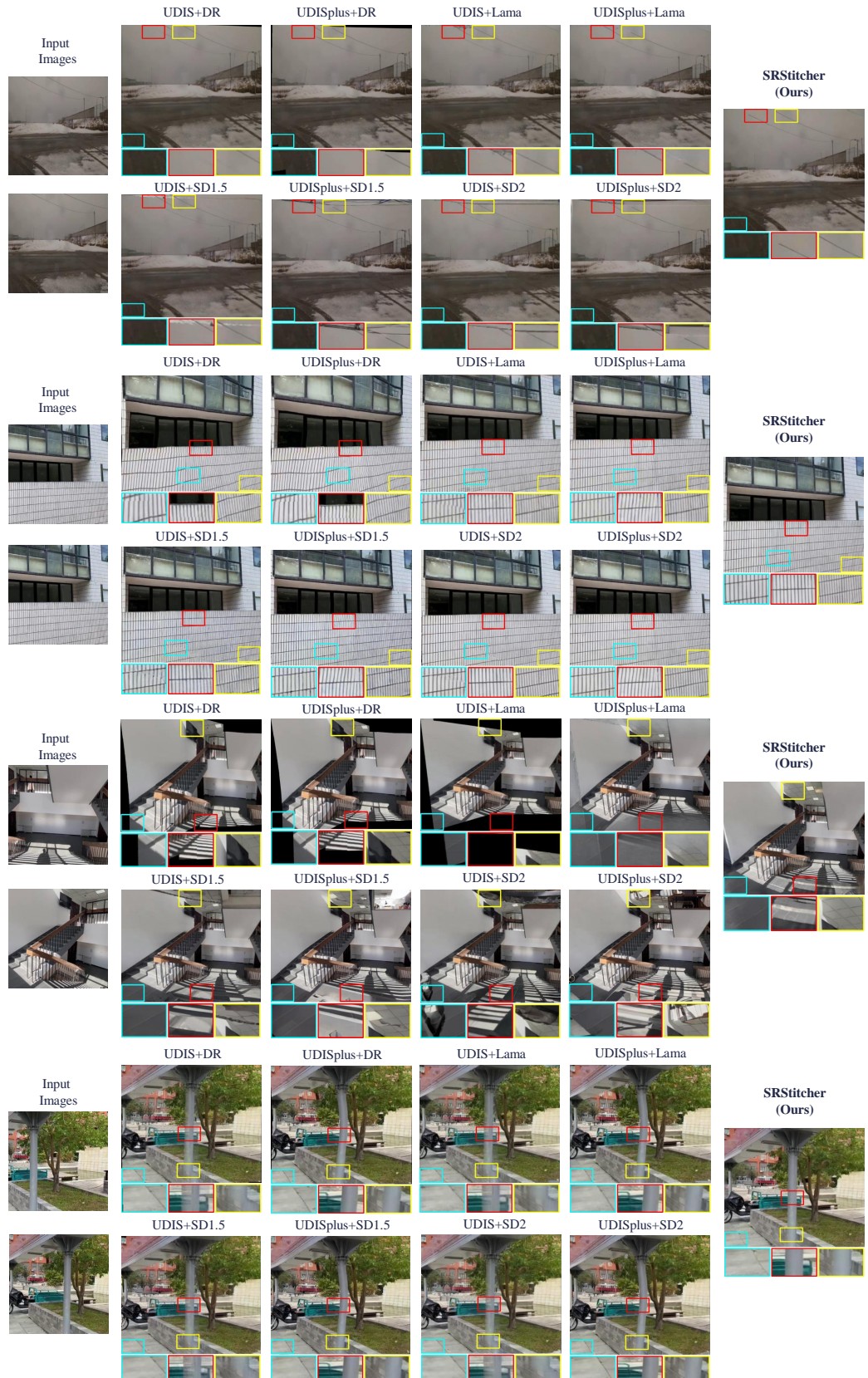

Figure 13: Evaluation results of the example in Figure 2 on all baselines.

## D.2 Additional hyper-parameters study results

**(1) The impact of $K_g$**

The $K_g$ in in Eq. 10 and Eq. 11 determines the intensity of the distance transform applied within the weighted masks. Despite the potential variations available in the type of distance transform (e.g., L1 vs. L2) and the kernel size, our empirical observations show that these modifications do not significantly impact the stitching results. Therefore, we set a commonly used value using an L2 distance and a kernel size of 3.

**(2) The impact of seed**

We show the impacts of different seeds in Figure 14. Our method produces more stable results with high quality. With different random seeds, Stable-Diffusion-v1-5-inpainting (SD1.5) [41] and Stable-Diffusion-v2-inpainting (SD2) [3] produce completely different abnormal contents. In contrast, our proposed method consistently demonstrates remarkable stability.

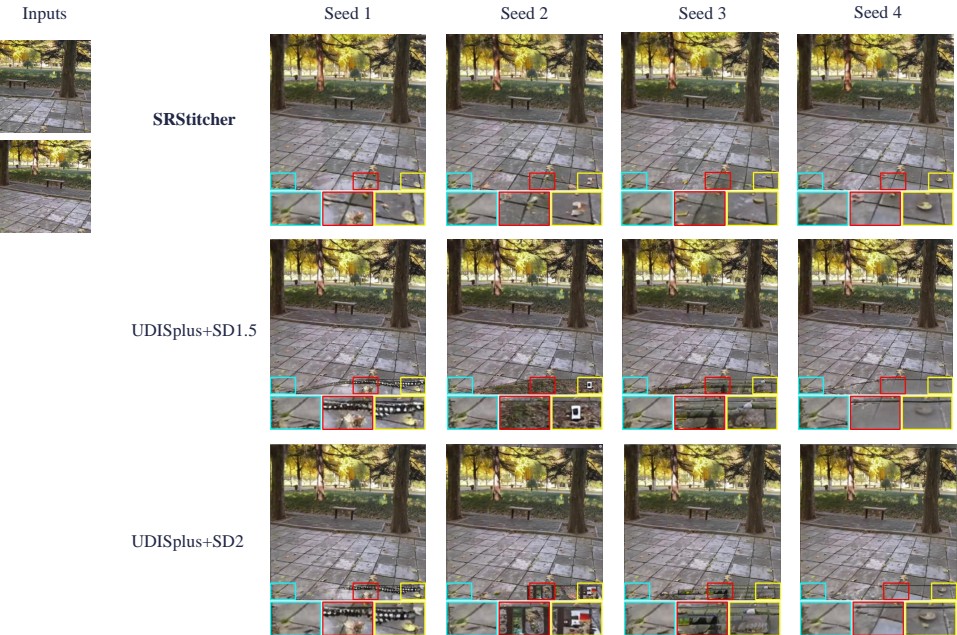

Figure 14: Ablation study of the seed.

**(3) The guidance scale and inference step**

The guidance scale and inference step are two classical parameters in diffusion models. The impact of adjusting these parameters has been extensively validated by previous research [48]. Consequently, this paper does not delve into selecting their values but instead adopts two commonly used settings: 7.5 and 50.

## D.3 Qualitative evaluation results of SRStitcher variants

This section presents the qualitative evaluation comparing SRStitcher variants based on various diffusion models. The version based on the Stable Diffusion 2 model [2] is designated as SRStitcher-S. Additionally, the implementation utilizing the Stable Diffusion 2 Unclip model [4] is termed SRStitcher-U. Finally, the implementation with Controlnet Inpainting model [54] is defined as SRStitcher-C.

The test results are shown in the Figure 15. The Stable-Diffusion-v2-inpainting model exhibits the best performance, which is the primary reason for selecting it as our base model. The Stable-Diffusion-2-1-Unclip, a fine-tuned model based on Stable-Diffusion V2.1, is chosen in an attempt to leverage its CLIP image embedding functionality. However, the structural integrity of the results

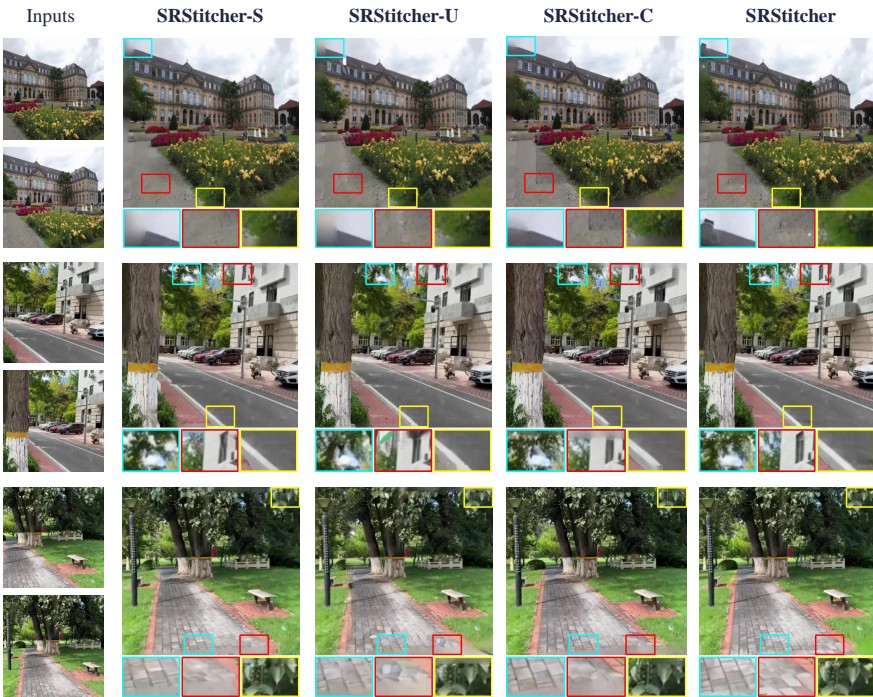

Figure 15: Qualitative evaluation results of SRStitcher variants.

generated by this model is significantly inferior to those of the other two models, likely due to compromises introduced during fine-tuning.

Notably, the performance of the SRStitcher-C based on ControlNet has exceeded our expectations. While the model does exhibit a more pronounced issue with local blurring, it demonstrates exceptional capability in preserving the original image information. In future work, should model fine-tuning be employed to further optimize the stitching effect, we may consider beginning our enhancements with the ControlNet model.

We provide the source code for all SRStitcher variants in the Supplementary Material. Therefore, we do not describe implementation of them here.

### D.4 Generalization on other datasets

In addition to the UIDS-D dataset, there are traditional datasets in the field of image stitching, such as APAPdataset [52] and REWdataset [26]. However, these datasets are very small, containing only dozens of images, which reduces their usefulness for meaningful comparative experiments. To demonstrate the effectiveness of our method on a broader spectrum of data, we present some experimental results on APAPdataset and REWdataset, as illustrated in Figure 16.

In addition, Figure 17 shows the experimental results comparing SRStither and other methods on the traditional datasets.

### D.5 Examples of local blurring

Here, we present examples of local blur and compare them with other baselines, as illustrated in Figure 18. We contend that occasional local blur is a tolerable side effect of our scheme, especially when weighed against the generation of significant anomalous content seen in other models. Future research could potentially address this issue by fine-tuning the model.

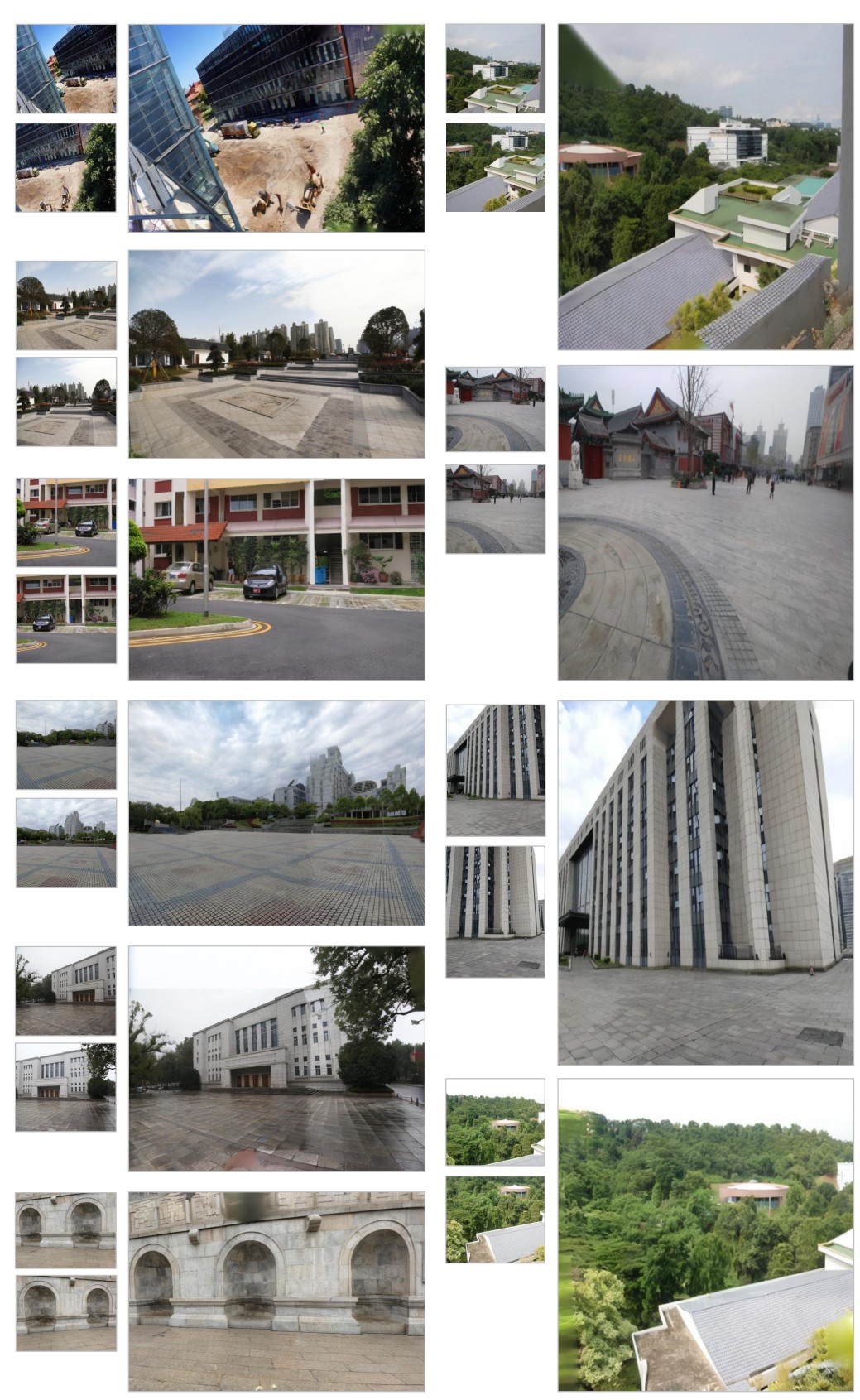

Figure 16: More results on traditional datasets APAPdataset [52] and REWdataset [26] by SRStitcher.

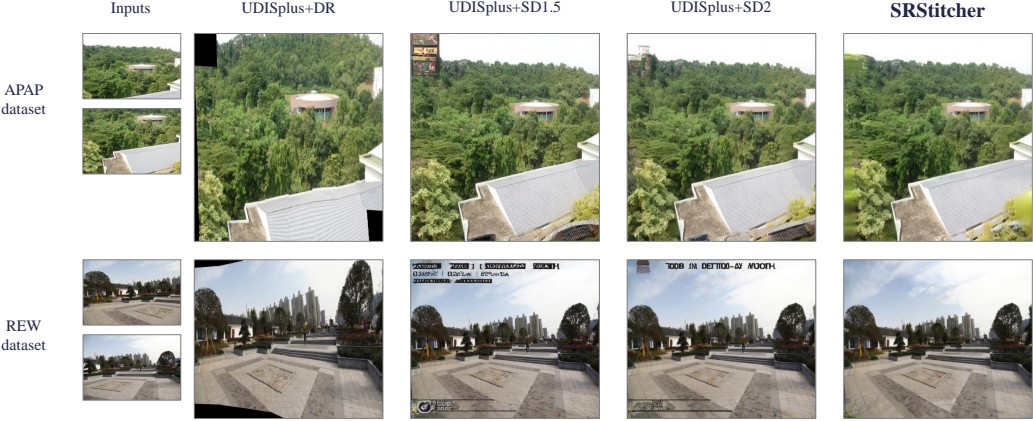

Figure 17: Comparison results on traditional datasets APAPdataset [52] and REWdataset [26].

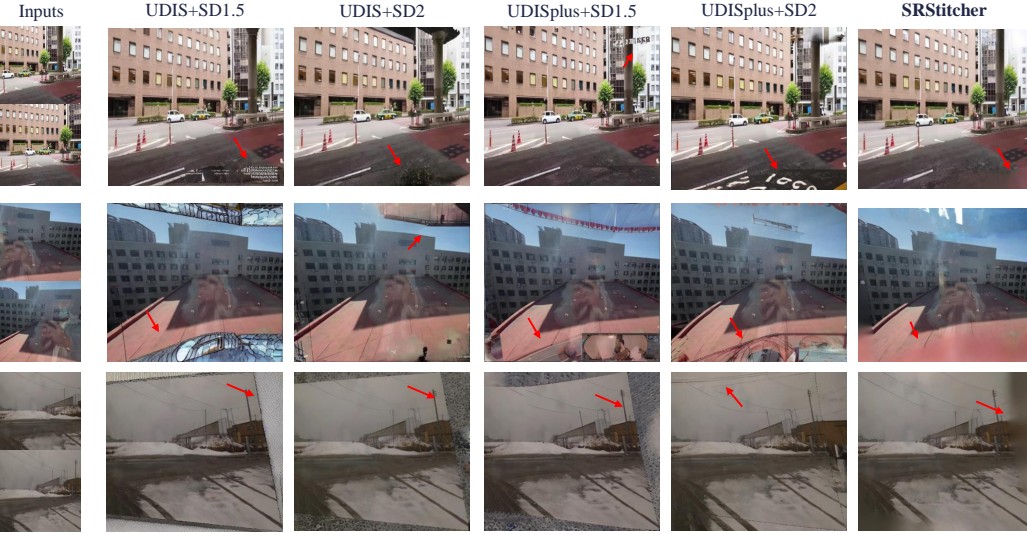

Figure 18: Examples of local blurring.

## D.6 Speed

Our solution requires only a single inference step, making it significantly faster than more complex models that require two inference steps, such as UDIS+SD1.5 to UDISplus+SD2. Although our method is slightly slower compared to UDIS+DR to UDISplus+Lama, our experimental results demonstrate that it substantially outperforms these methods regarding stitched image quality, robustness, and generalization. Given these advantages, the minor sacrifice in speed is justifiable. In particular, even without acceleration optimizations like TensorRT [37], our method achieves an average processing speed of 27 it/s on an NVIDIA 4090 GPU, which is sufficient for real-time performance.

## E Broader impact

This paper aims to introduce a novel image stitching pipeline design that integrates large-scale models into the image stitching process. Unlike previous diffusion model-based image stitching methods, our method does not require training or task-specific supervised datasets. This significantly lowers the implementation threshold, facilitating broader adoption of the method and encouraging more researchers to contribute and enhance this work. One potential negative impact of our method is its

ability to make minor modifications to stitched images. This capability could be misused in visual fraud tasks, particularly in the context of surveillance videos.

## F   Reproducibility

We have made significant efforts to ensure the reproducibility of our method. The code and details of the SRStitcher, Stitcher variants, and CCS metric are all uploaded in the Supplementary Material.

