# OpenReview forum: "Reconstructing the Image Stitching Pipeline: Integrating Fusion and Rectangling into a Unified Inpainting Model"
_NeurIPS.cc/2024/Conference — NeurIPS 2024 poster_

### Official Review · Reviewer_yn5D · 2024-07-08

**Soundness:** 3
**Presentation:** 3
**Contribution:** 2
**Rating:** 5
**Confidence:** 5

**Summary:**

This work proposes to integrate the fusion and rectangling of image stitching into a unified inpainting model. In particular, the weighted masks are designed to guide the reverse process in a pre-trained large-scale diffusion model, which implements this integrated inpainting task in a single inference. Extensive experiments demonstrate the interpretability and generalization capabilities of the proposed unified model.

**Strengths:**

+ The motivation for reconstructing the image stitching pipeline into a unified model is sound and clear.
+ Some discussions on image stitching are insightful. For example, in the Introduction, the authors claimed that "To address the error propagation problem, we identify image fusion as the key point for improvement". I agree it is an accurate point to inspire this work.
+ The reviews of previous methods are detailed and comprehensive (in Tab. 1 and Section B in the Appendix), which clearly shows their limitations and potential improvements.

**Weaknesses:**

- While this work is well-motivated, the proposed contributions seem to be weak for me. For instance, the authors proposed a weighted mask to guide the reverse process in the diffusion model. However, the current presentation regarding this contribution looks like an experimental trick rather than a technical proposal.
- The contribution of the proposed unified inpainting model (Section 3.1) is also expected to be highlighted. It is still ambiguous how the fusion part and rectangling part are merged into a unified model. I believe this module is much more important than applying cutting-edge generative models.
- This work leverages a **generation** view to address the **reconstruction** problem. Such an interesting trend can be noticed in the recent works from other research regions as well. However, how to balance the generation and reconstruction in the image stitching problem remains unclear. I believe the users may prefer accurate reconstruction instead of introducing novel content (sometimes artifacts). Moreover, some previous works also proposed to fill the irregular boundaries of images or holes by warping with generative models, such as "Free View Synthesis (ECCV'2020)", "Towards complete scene and regular shape for distortion rectification by curve-aware extrapolation (ICCV'2021)", "iNVS: Repurposing Diffusion Inpainters for Novel View Synthesis (SIGGRAPH Asia)", etc. More discussions are expected to be provided.
- There are some typos in the paper: Line 49: "Therefore, We question". The authors are suggested to further polish this work.

**Questions:**

- Can the proposed unified model be extended into other research areas?

**Limitations:**

- The proposed unified inpainting model seems to be impractical compared to previous reconstruction-based solutions.

---

> ### Author Rebuttal · Authors · 2024-08-05
>
> We greatly appreciate your careful assessment of our work. Please see our responses below.
>
> ## Weaknesses
>
> We start with weakness 3, and then discuss weaknesses 1, 2 and 4.
>
> **W3: The reconstruction-based image fusion model has significant limitations and has fallen into the development bottleneck.**
>
> *Users may prefer accurate reconstruction, but only if the reconstruction is truly accurate.*
>
> First reconstruction-based image fusion model is a UNet-like model proposed by VFISnet [A] in 2020. The subsequent classical learning-based image stitching works [B, C, D] generally follow this model structure, and there are few structural breakthrough innovations.
>
> Now, the SOTA reconstruction-based model is UDIS [C]. However, it is mentioned in UDIS++ that UDIS has significant defects, and UDIS may introduce blurred parallax regions in the fusion image with large parallax. Therefore, UDIS++[D] focused on finding a suitable soft seam and ultimately gave up improving the reconstruction-based fusion model.
>
> Based on the above discussion, in image stitching, the reconstruction-based image fusion model research has not been a fundamental breakthrough for many years, and its SOTA method UDIS still has apparent defects. Therefore, we propose another idea to explore a new direction of image fusion based on the generative model.
>
> A single research perspective limits the breadth of the development of a research field. We believe our method provides a new research perspective for the fusion problem in image stitching.
>
> **Related work of irregular boundaries** Thank you for reminding us of the work related to irregular boundaries. We will add the discussion about these works in the related work.
>
> **W1: The construction of the weighted mask has the theoretical basis.**
>
> Due to the space limitation of the main paper, we put the detailed theoretical basis and design principle in Appendix A "More Details of SRStitcher".
>
> To address this weakness, we thoroughly rewrote and supplemented Weighted masks parts in Section 3.2 Weighted mask guided reverse process in the new manuscript, as shown in the gray box.
>
> Weighted inpainting mask is inspired by the suffix principle [E]. This principle allows for the customization of the variation for each pixel or image region during the reverse process. We extend this concept by applying it to the diffusion-based Inpainting model and incorporating the requirements of image stitching task to construct weighted masks.
>
> > The weighted inpainting mask, as described in Eq. 11, is inspired by the suffix principle [E]. During the reverse process, weighted inpainting mask is mapped into multiple sub-masks to define the modified regions at each step $t$. ....This gradual modification method facilitates a more seamless blending of the inpainting content with the original image content.
>
> However, the suffix principle is a scheme to smooth the transition between the inpainting and non-inpainting areas, which cannot perceive the original content. Inspired by the attention map, we design the weighted initial mask, which assigns different fidelity to each pixel in the original image.
>
> > The weighted initial mask assigns different fidelity levels to each pixel of the fusion image, determining how much to modify each pixel based on its fidelity during the reverse process. The formula of weighted initial mask is given by Eq. 10, which is composed of two parts. The left part determines the fidelity levels of pixels in $M_{seam}(x,y)$ region, and the right part determines the fidelity levels of pixels in $M_{content}(x,y)$ region.
>
> Weighted masks are a further extension of the suffix principle, which discusses how to achieve more fine-grained and content-fidelity inpainting control and provides some technical contributions to image stitching and the control of diffusion-based inpainting process.
>
> **W2: We believe that the challenge of unified models is not in definition but in implementation.**
>
> Our unified model definition is straightforward and intuitive (Eq. 8). The real difficulty lies in finding a suitable model to implement this unified problem efficiently. The image stitching is a field where data is extremely scarce. All datasets used in this paper are unlabeled, which makes it difficult to train the model.
>
> *So, our first task is not to design a theoretically elegant model, but to design a practically feasible model.*
>
> Without prior knowledge provided by large-scale generative model, corrected registration errors and unsupervised rectangling are a struggle to succeed. Therefore, in the description of the main paper, we emphasize how to be compatible with the structure of existing large-scale generative models in the unified problem definition. In doing so, our method can introduce large-scale models into image stitching, which opens a new research direction for image stitching with extreme data shortage.
>
> **W4: Thank you for your careful check. We have re-checked the paper and revised all typos.**
>
> ## Questions
>
> Our unified model is specifically designed for image stitching pipelines. Given that we have yet to conduct a systematic study of the application of the model to other domains, we are careful to avoid asserting its universality in other domains. The core of our technique lies in its fine processing ability of image content to correct abnormal content while preserving the original content, which is potentially valuable for applications in multiple low-level vision domains, such as image restoration, artifact removal, and low-light enhancement.
>
> ## Limitations
>
> Answered, see W3.
>
> **references**
>
> [A] A view-free image stitching network based on global homography, 2020.
>
> [B] Learning edge-preserved image stitching from multi-scale deep homography, 2022.
>
> [C] Unsupervised deep image stitching: Reconstructing stitched features to images, 2021.
>
> [D] Parallax-tolerant unsupervised deep image stitching, 2023.
>
> [E] Differential diffusion: Giving each pixel its strength, 2023.

---

> > ### Comment · Reviewer_yn5D · 2024-08-14
> >
> > Thanks for the detailed rebuttal, which addressed most of my concerns. I would like to increase my rating. The updated clarifications and experiments are expected to be presented in the final version.

---

### Official Review · Reviewer_pLdt · 2024-07-10

**Soundness:** 3
**Presentation:** 3
**Contribution:** 2
**Rating:** 6
**Confidence:** 4

**Summary:**

This paper proposes an image stitching algorithm that unifies fusion and rectanguling stages of a conventional pipeline with an image inpainting diffusion model applied with a progressive reverse process guided by weighted masks. Their reverse algorithm progressively inpaints seam regions by gradually increasing the size of soft mask holes, while keep outpainting rectanguling regions with the same hard mask holes. The proposed work is compared with previous works by keeping the registration and fusion stages with UDIS/UDIS++ while altering with various models for a rectangling stage.

**Strengths:**

In their setup of experiments, theirs generally showed higher quality quantitatively and quantitatively than compared methods.

The proposed reverse process results in a high quality generation, suitable to inpaint and outpaint the seam and rectangling regions naturally.

**Weaknesses:**

It is unclear if the repeated reverse process with gradually dilating mask holes is really beneficial than a single reverse process guided by a fixed mask. The repeated application of the re-noising + de-noising step makes the algorithm much slower.

Considering the seam regions are usually narrow while the rectanguling regions could be quite wide, the progressive soft mask dilation during the progressive reverse process may be more useful with the rectanguling masks than the seam masks.

**Questions:**

Comparing with a case of a single reverse process guided by a fixed mask would make the paper more convincing.

Comparing with a case that applies progressive mask dilation of the rectanguling mask holes would be useful.

**Limitations:**

Technical limitations are properly addressed. No serious potential negative societal impact is expected.

---

> ### Author Rebuttal · Authors · 2024-08-05
>
> Thank you for the thoughtful review of our work. Please see below for the responses.
>
> ## Weaknesses
>
> Please see the newly added PDF in Global Response. We have provided the Rebuttal Fig. 3 and Rebuttal Fig. 4 to explain your confusion.
>
> **W1: Gradually dilating mask holes in seam regions are based on content preservation consideration.**
>
> Imagine that there is a painter, and this painter has an unfinished painting(Coarse fusion image). There are two things the painter needs to finish on this painting:
>
> 1. The painter is not satisfied with the finished part of the painting and needs to modify it (seam regions).
> 2. There are still parts of the painting that are not painted (rectanguling regions).
>
> Let us do the first thing: if this painter smears out everything in the seam region at once (fixed mask), the modified image differs significantly from the original content, see Rebuttal Fig. 3(a) red box.
>
> So this painter needs to modify it slowly, a little at each time (step $t$), so that the painter can refer to what is still in the seam regions to ensure that the modified content still has the original content, see Rebuttal Fig. 3(b) red box. This is why we use Gradually dilating mask holes in seam regions.
>
> **W2: Using gradually dilating mask holes in rectangling regions is disappointing.**
>
> Let us do the second thing: the painter wants to paint the rectangling regions using the same strategy as seam regions. However, there is nothing in this region. When the painter starts drawing from a small mask, the painter finds the information near the mask is insufficient.
>
> One side of the mask has the content, and the other has no content (Rebuttal Fig. 4(b)). Alternatively, one side of the mask has no content, and the other is directly out of the image (Rebuttal Fig. 4(a)).
>
> The painter must smooth the contents of both sides of the mask with each step, and smoothing the image without content introduces blurring noise.
>
> If the painter were a human, he would know that he only needed to consider the pixels with content. Unfortunately, the painter is a robot whose execution program is inflexible.
>
> Therefore, in rectangling regions, we use fixed mask. One side of the mask has content, and the other is directly out of the image.  Our smart robot painter knows that there is no need for smoothing the content beyond the image.
>
> **Speed: Our method does not increase the inference time.**
>
> Our design does not increase the computational complexity of the model, and the re-noising + de-noising step is the standard process of the diffusion model. The original diffusion model still needs multi-step sampling without our method [A]. We achieve fine-grained control for local adjustment of different image regions by using weighted masks in the reverse process. This approach keeps the basic computational structure of the model the same.
>
> Thus, the inference time of the model with our method remains relatively consistent compared to the original model.
>
>
> ## Questions
>
> **Q1: We provide a case of the reverse process guided by a fixed mask in the new pdf.**
>
> It is shown in Rebuttal Figure 3. There is no single reverse process. The reverse process of the diffusion model is multi-step [A].
>
> **Q2: We provide a case that applies progressive mask dilation of the rectanguling mask holes in the new pdf.**
>
> It is shown in Rebuttal Figure 4.
>
> **references**
>
> [A] Denoising diffusion probabilistic models, 2020.

---

> ### Comment · Reviewer_pLdt · 2024-08-12
>
> Thank you for your rebuttal. It clarifies the need of different masking strateges for the seam and rectangling regions, and the preservation of computational complexity for the inference. Therefore, my major concerns have been resolved.
>
> I find that the proposed scheme, which converts conventional multiple stages for a stiching problem into unified one by adapting diffusion based inpainting models, holds certain values suitable for the venue. Existing inpatining models would not be able to achieve this goal without applying the proposed algorithm.
>
> I would like to update the final rating from from BA to WA.

---

### Official Review · Reviewer_m57w · 2024-07-13

**Soundness:** 3
**Presentation:** 3
**Contribution:** 3
**Rating:** 6
**Confidence:** 4

**Summary:**

The paper introduces SRStitcher, a novel method that integrates the fusion and rectangling stages of the image stitching pipeline into a unified inpainting model using a pre-trained large-scale diffusion model, eliminating the need for additional training. This approach addresses the issue of error propagation in traditional pipelines, offering a streamlined and robust solution. Strengths of the method include improved performance in image quality and content consistency, as well as robustness to registration errors. The experimental results are extensive, providing both quantitative and qualitative evidence of SRStitcher's superiority over existing state-of-the-art methods.

**Strengths:**

SRStitcher's primary strength lies in its innovative integration of the fusion and rectangling stages into a unified inpainting model, which addresses the long-standing issue of error propagation in traditional image stitching pipelines. By leveraging a pre-trained large-scale diffusion model, SRStitcher eliminates the need for stage-specific training, thereby simplifying the pipeline and enhancing its robustness. This approach ensures superior performance in handling registration errors, which are typically propagated and amplified in multi-stage pipelines. Additionally, the use of weighted masks to guide the inpainting process allows for precise control over inpainting intensity, significantly improving image quality and content consistency. These advancements address the limitations of existing methods, which often struggle with the independent optimization of each stage and the associated parameter tuning challenges. The extensive experimental results, including both quantitative metrics and qualitative assessments, robustly demonstrate SRStitcher's superiority over state-of-the-art methods, showcasing its ability to produce high-quality stitched images with greater stability and fewer artifacts.

**Weaknesses:**

While SRStitcher simplifies the image stitching pipeline by integrating the fusion and rectangling stages into a unified inpainting model, its technical novelty is questionable. The approach primarily combines existing technologies—pre-trained diffusion models and weighted masks—rather than introducing fundamentally new methodologies. The integration, while effective, does not inherently surpass the capabilities of current state-of-the-art techniques used separately for fusion and rectangling. For SRStitcher to be considered truly novel, it should achieve technical goals that were unattainable with the two processes handled independently. As it stands, the method appears to be more of a consolidation of existing practices rather than a groundbreaking innovation, merely reorganizing the workflow without providing substantial new capabilities or overcoming significant limitations of the previous approaches.

**Questions:**

- Technical Novelty: How does SRStitcher fundamentally advance the field of image stitching beyond merely integrating existing fusion and rectangling processes into a unified model? Can the authors provide more evidence of novel technical contributions?

- Performance Comparison: While the paper claims superior performance, how does SRStitcher specifically outperform existing state-of-the-art methods in scenarios with extreme registration errors(large parallax)? Are there any edge cases where SRStitcher struggles compared to traditional methods?

**Limitations:**

The authors have adequately addressed the limitations of their work and there appear to be no significant issues with the broader societal impacts. They have demonstrated transparency and responsibility in discussing the constraints and potential improvements of SRStitcher. Overall, their approach seems robust and well-considered, with no apparent concerns regarding its application or societal implications.

---

> ### Author Rebuttal · Authors · 2024-08-04
>
> We thank the reviewer for the comments and careful reading. Please see our responses below.
>
> ## Weaknesses
>
> We would like to clarify that SRStitcher achieves technical goals that were unattainable with the two processes handled independently, especially in image rectangling.
>
> **1. SRStitcher improves the robustness of the fusion technique**
>
> The image fusion method can be broadly categorized into two types: reconstruction-based method and seam-based fusion method. The SOTA  reconstruction-based model is UDIS[A], which has been observed to introduce blurring parallax regions, particularly in scenes with large parallax. This issue is thoroughly documented in UDIS++[B]. To address this, the SOTA seam-based model UDIS++ attempts to improve the fusion image quality by the soft seam. However, in scenarios with registration errors between images, a perfect seam does not exist. In such scenarios, UDIS++ forces the image distortion to "create" a perfect seam, which results in the distortion of the pillar as illustrated in Figure 1② of our paper.
>
> To address the aforementioned issues, we propose a new solution for image fusion that uses the inpainting-based method to smooth the image. In contrast to reconstruction-based methods, SRStitcher does not introduce blurred regions when handling large parallax scenes. Furthermore, in comparison to seam-based methods, SRStitcher does not depend on the existence of perfect seams in the image. Consequently, SRStitcher exhibits greater robustness in dealing with registration errors compared to previous methods.
>
> **2. SRStitcher  is the first rectangling solution that does not require supervised data and has higher generalization performance**
>
> SRStitcher represents a major breakthrough for the rectangling problem. Existing rectangling methods [C, D, E] are all supervised learning methods and require labeled datasets. And the only labeled dataset for image rectangling is DIR-D[C]. DIR-D is an ideal dataset that excluded some challenging scenes during its production process, and its scale is relatively small. Models trained on DIR-D have limited generalization ability, as evidenced by zero-shot experimental results of DeepRectangling.
>
> Our solution is the first rectangling method that does not require supervised data. It generalizes much better than the SOTA rectangling model DeepRectangling,  due to the prior knowledge from large-scale pre-trained models.
>
> We appreciate the reviewer's comments, which helped us identify the areas for improvement in the writing of our contributions section. We modify the contributions in the Introduction section as follows:
>
> > The main contributions of this paper are:
> (1) We propose SRStitcher, which reformulates the problem definitions of the fusion and rectangling stages to construct a more streamlined and robust image stitching pipeline.
> (2) SRStitcher is the first to introduce the concept of inpainting to address the image fusion problem. It incorporates prior knowledge from large-scale pre-trained models into the image stitching pipeline, enhancing the robustness of image fusion against registration errors.
> (3) Without additional fine-tuning or supervision, SRStitcher improves the generalization of the rectangling method in the zero-shot scenario, opening up new possibilities for unsupervised image rectangling research.
> (4) We conduct extensive experiments to verify the interpretability and generalization of the proposed unified model. The results show that SRStitcher significantly outperforms the state-of-the-art methods in both quantitative and qualitative evaluations.
>
> ## Questions
>
> **Q1: SRStitcher has three novel technical contributions**
>
> 1. SRStitcher first proposed the inpainting-based fusion method, a new thinking direction.
>
> 2. SRStitcher first incorporates prior knowledge from large-scale pre-trained models into the image stitching pipeline.
>
> 3. Weighted masks are not existing techniques, we design them to get a more refined inpainting control.
>
> Thanks to the reviewer' suggestions, we have revised the description in the Introduction section to emphasize the significant contributions of our method in both fusion and rectangling techniques.
>
> **Q2: We provide a case for an extreme registration error scenario in the Rebuttal Fig. 2**
>
> Please see the newly added PDF in Global Response for Rebuttal Fig. 2.
>
> In this scene, due to the large parallax and complex floor tile texture, other comparison fusion methods exhibit noticeable misalignment and artifacts. SRStitcher addresses these issues using an inpainting-based method. Although SRStitcher cannot guarantee perfect results in the presence of such significant registration errors, it markedly outperforms previous methods in terms of tile continuity.
>
> According to the definition of $K_s$ (Lines 187), our method adaptively increases the width of $M_{seam}(x,y)$ in large parallax scenarios. Therefore, our method performs effectively in large parallax.
>
> Edge cases exist. Specifically, in "small parallax scenes with substantial color differences between the stitched images", our method may exhibit more pronounced stitching seams compared to existing methods such as UDIS and UDIS++.  Because, the current hyper-parameter settings are relatively conservative, resulting in a relatively small value of $K_s$ in small parallax scenes. The width of $M_{seam}(x,y)$ is insufficient to provide enough space for color difference smoothing. We think that this issue can be mitigated by pre-calculating the color differences and design a more flexible hyper-parameter setting method.
>
> **references**
>
> [A] Unsupervised deep image stitching: Reconstructing stitched features to images, 2021.
>
> [B] Parallax-tolerant unsupervised deep image stitching, 2023.
>
> [C] Deep rectangling for image stitching: A learning baseline, 2022.
>
> [D] Recdiffusion: Rectangling for image stitching with diffusion models, 2024.
>
> [E] RectanglingGAN: Deep rectangling for stitched image via image inpainting, 2024.

---

### Official Review · Reviewer_GE1w · 2024-07-14

**Soundness:** 2
**Presentation:** 2
**Contribution:** 2
**Rating:** 6
**Confidence:** 4

**Summary:**

This paper tried to integrate the fusion and rectangling stages in image stitching into a unified model. More concretely, a special fusion, a rectanlging step, and a mask-guided diffusion model are gathered to implement stitching-customized image inpainting, especially for the irregular boundaries. It is worth mentioning that, the inpainting model uses the pre-trained model and requires no more fine-tuning. To evaluate the proposed method, the authors designed a quantitative metric named CCS and conducted extensive experiments.

**Strengths:**

1. The concept of integrating multiple stages of image stitching into a single stage is novel and promising.
2. The experiments are abundant and convincing.
3. The authors leverage a pre-trained model to implement stitching-customized inpainting without any extra training.

**Weaknesses:**

1. The so-called unified model consists of several steps including a specially-designed fusion step, a rectangling step, and a inpainting step. I don’t think this multi-step design can be regarded as a unified model.
2. I am skeptical of whether the inpainting model is meaningful to image stitching. As claimed in the DeepRectangling paper, they abandoned the inpainting model because they think it may introduce some contents that are far from the reality. The manuscript seems to ignore the problem. And I do not think the proposed method can address this issue in the so-called unified inpainting model.
3. The inpainting results are still not perfect, as illustrated in the image boundaries of Fig. 14.
4. The fusion step of Eq. 4 is special. What’s the motivation for that?

**Questions:**

My concern lies in my second weakness. Will this inpainting model for irregular boundaries be meaningful? I have another idea for the inpainting model of image stitching. The inpainting model should not concentrate on the boundaries but on the regions where the artifacts and distortion are produced. Let’s put it in this way. The registration stage may introduce artifacts or distortion. So can we eliminate these issues through an inpainting model? It may be more meaningful to locate these regions, mask them, and then inpaint them. The inpainting process can be implemented with the guidance of original contents, thus contributing to a reliable completion model. This is just a simple discussion. But from my perspective, I cannot figure out the meaning of the proposed inpainting model.

**Limitations:**

No limitation is mentioned in the manuscript.

---

> ### Author Rebuttal · Authors · 2024-08-04
>
> We thank you for raising these issues and your comments. Please see below for the responses.
>
> ## Weaknesses
>
> We start with weakness 2, which is most concerned by the reviewer, and then discuss weaknesses 1, 3 and 4.
>
>  **W2: DeepRectangling's experimental results of inpainting models are outdated.**
>
> The DeepRectangling paper claimed that:
>
> > Nevertheless, there is currently no work to design a mask for irregular boundaries in image stitching, and even SOTA completion works [26, 28] show unsatisfying performance (Fig.1d) when processing the stitched images.
>
> However, it is important to note that DeepRectangling paper was published in 2022, and its references 26[A] and 28[B], were completed in 2021 and 2019, respectively. Notably, since 2022, there has been substantial progress in diffusion-based inpainting methods[C, D, E], which demonstrate far superior performance in terms of image quality and semantic coherence. Therefore,  [A, B] in DeepRectangling paper are not SOTA now.
>
> Also, a recent work[F], published in June 2024 (and therefore not cited in this paper), has begun exploring inpainting-based methods to address the rectangling problem, yielding promising results. Thus, the potential and value of the inpainting-based method for rectangling problem should not be denied based only on the results in DeepRectangling paper.
>
> To address the unrealistic content problem mentioned in the DeepRectangling paper, we propose an innovative solution, based on the weighted initial mask and coarse rectangling, that effectively controls abnormal content. The unrealistic content problem has been successfully resolved in our method.
>
> **W1: The unified model is clearly defined in the paper by Eq. 8.**
>
> In Section 3.1 Unified inpainting model,  we first define the image fusion problem(Eq. 4), and then define the image rectangling problem(Eq. 6). Finally, we explain how these two originally separate problems can be abstracted into one unified problem that can be solved by one single learning-based model (Eq. 8). This is not a "multi-step design", but rather a conceptual process of integrating two individual problems into a unified framework.
>
> **W3: No method can perfectly stitch the images in Fig. 14.**
>
> In the newly added PDF in Global Response, we provide the Rebuttal Fig. 1 to show the performance comparison of our method with UDISplus+DR, UDISplus+SD1.5, and UDISplus+SD2 under the scenes in Fig. 14.
>
> The results show that DeepRectangling causes distortion in the overall structure of the image, while both SD1.5 and SD2 introduce content in the upper-left corner of the image that is not present in the original. In contrast, our method exhibits only minor issues with edge clarity.
>
> Local blurring is a limitation of our method, which is discussed in the 5 Discussion and conclusion of the main paper (Lines 249-254). This blurring results from a trade-off to reduce the likelihood of generating abnormal content. The presented limitations provide directions and possibilities for future research. So, we do not shy away from our imperfections in Fig. 14, they do not imply that our method is inferior to other methods.
>
> **W4: The motivations of Eq. 4 are in the text directly above this equation and Introduction section.**
>
> The text directly above Eq. 4 (Lines 90-93) : *"Precisely, as shown in Eq. 2, the distortion degree of $I_{wl}(x, y)$ is relatively low because it involves only minor warping based on $\texttt{I}$. This means that even in the presence of registration errors, $I_{wl}(x, y)$ does not introduce large-scale distortions. Therefore, we propose to construct a coarse fusion image $I_{CF} (x, y)$ via Eq. 4."*
>
> Also, in the Introduction section (Lines 44-46) : *"We propose to reformulate the fusion problem by overlaying the less distorted aligned image over the more distorted one, and inpainting the seam area between the images to correct the inappropriate image content."*
>
> ## Questions
>
> **The "another idea" is the core idea of our paper.**
>
> In the Questions, the reviewer proposes an "another idea". We believe it is a super excellent idea, because this "another idea" is highly consistent with the core idea of our paper introduced in the Introduction section (Lines 41-44) : *"Therefore, we reconsider the problem definition of the fusion challenge and hypothesize that: By determining the appropriate modification region and introducing an inpainting model with strong generalization ability, the abnormal image content caused by registration error can be effectively corrected."*
>
> We have detailed how our core idea is implemented in the paper. Therefore, this so-called "another idea" proposed by the reviewer has been realized in our paper.
>
> **Overall, our unified inpainting model is meaningful.**
>
> 1.  Our response to "weakness2" has proved that the results from DeepRectangling do not indicate that inpainting-based methods are ineffective for rectangling problem.
>
> 2. The so-called "another idea" proposed by the reviewer has been implemented in our paper.
>
> 3. Experiments  (Fig. 2 and Rebuttal Fig. 1)  show that our method significantly outperforms DeepRectangling in the ability to preserve the original structure of the fusion image.
>
> Based on the aforementioned arguments, we are confident that the unified inpainting model is effective and has a solid theoretical foundation and significant experimental results.
>
> ## Limitations
>
> We mentioned the limitations in section "5 Discussion and conclusion" of the manuscript.
>
> **references**
>
> [A] Resolution-robust large mask inpainting with fourier convolutions, 2021.
>
> [B] Boundless: Generative adversarial networks for image extension, 2019.
>
> [C] Repaint: Inpainting using denoising diffusion probabilistic models, 2022.
>
> [D] Palette: Image-to-image diffusion models, 2022.
>
> [E] Smartbrush: Text and shape guided object inpainting with diffusion model, 2023.
>
> [F] RectanglingGAN: Deep rectangling for stitched image via image inpainting, 2024.

---

> > ### Comment · Reviewer_GE1w · 2024-08-14
> >
> > My concerns are addressed in the rebuttal and I would like to raise my evaluation.

---

### Author Rebuttal · Authors · 2024-08-05

We thank all the reviewers for their careful comments and agreement with our motivation. Here, We address some of the concerns shared by multiple reviewers and upload a PDF with rebuttal figures.

**1. We clarified key contributions**

Due to the previous manuscript's limited space, our contributions are not clear enough, leading reviewers to underestimate our work's technical contributions to the image stitching. Here, we revised the contributions at the end of Introduction section as follows:

>The main contributions of this paper are:
>
>1. We propose SRStitcher, which reformulates the problem definitions of the fusion and rectangling stages to construct a more streamlined and robust image stitching pipeline.
>
 >2. SRStitcher is the first to introduce the concept of inpainting to address the image fusion problem. It incorporates prior knowledge from large-scale pre-trained models into the image stitching pipeline, enhancing the robustness of image fusion against registration errors.
>
>3.  Without additional fine-tuning or supervision, SRStitcher improves the generalization of the rectangling method in the zero-shot scenario, opening up new possibilities for unsupervised image rectangling research.
>
 >4. We conduct extensive experiments to verify the interpretability and generalization of the proposed unified model. The results show that SRStitcher significantly outperforms the state-of-the-art methods in both quantitative and qualitative evaluations.

**2. Feasibility of the unified inpainting model**

1. Feasibility of applying the Inpainting model to the fusion problem

At present, image fusion models are mainly divided into two categories: reconstruction-based and seam-based. SOTA reconstruction-based model UDIS [A] has been shown to introduce blurring parallax regions when dealing with large parallax inevitably. When there are registration errors between images, SOTA seam-based method UDIS++ [B] forcibly searches for seams, which leads to severe image distortion, as shown in Fig.1 ② in our paper.

To solve the above problems, we propose a new solution to the image fusion problem based on inpainting model. Compared with reconstruction-based methods, our method does not introduce blurred regions when dealing with large parallax scenes. In contrast to seam-based methods, our method does not rely on the assumption that perfect seams exist in the fusion image. Therefore, our inpainting-based method improves the fusion robustness for registration errors.

Our method needs to paint some image pixels, and we use the seam mask to limit the scope of modification strictly. Secondly, we introduce weighted masks to control the intensity of modification, ensuring the image's semantic consistency before and after modification. Extensive experiments confirm the effectiveness of our method. Therefore, we firmly believe that the proposed method is practical for the fusion problem.

2. Feasibility of applying the Inpainting model to the rectangling problem

One reviewer questioned the feasibility of applying the inpainting model to the rectangling problem based on the DeepRectangling paper. However, the DeepRectangling [C] paper was published in 2022, and diffusion-based inpainting methods have substantially progressed after 2022. Therefore, the possibility of applying the current diffusion-based Inpainting model in the image rectangling problem can not be denied based only on the conclusion of DeepRectangling. Also, RectanglingGAN [D], a paper published in June 2024, verifies the feasibility of the inpainting model for the rectangling problem.

Our method adopts the inpainting-based model to solve the rectangling problem. To deal with abnormal content generation, we design a combination strategy of coarse rectangling and weighted initial mask, which has solved the concern proposed by DeepRectangling. Experiments (Fig. 2 and Rebuttal Fig. 1) show that our method significantly outperforms DeepRectangling in the ability to preserve the original structure of the fusion image. Therefore, we firmly believe that the proposed method is practical for the rectangling problem.

**Improvement introduction:** Due to the space limitation, the Introduction section of the previous paper does not discuss in detail the inherent limitations of the existing fusion and rectangling methods and the breakthrough contribution of our method to them. We realize that it may lead to the reviewers underestimating the contribution of our method. Therefore, we have integrated the above discussion into our Introduction section in the revised paper.

**3. Controllability of generative model**

Some reviewers expressed concern that generative models might introduce uncontrollable content. We believe that the controllability of generative models is an optimistic research direction.

In recent years, the study of controllability of generative models has made remarkable progress and produced many widely influential works. For us, an important insight is provided by the work of Diff-Plugin[E], which verifies that large-scale pre-trained diffusion models and lightweight plugin networks can effectively handle low-level tasks in various visual domains, including de-rain, de-fog, and low-light enhancement while maintaining high-fidelity content consistency. Diff-Plugin confirms the ability of generative models in terms of content fidelity, giving us confidence to SRStitcher.

Therefore, we hold a positive attitude toward the controllability of generative models and believe that the research prospects in this area are up and coming.

**references**

[A] Unsupervised deep image stitching: Reconstructing stitched features to images, 2021.

[B] Parallax-tolerant unsupervised deep image stitching, 2023.

[C] Deep rectangling for image stitching: A learning baseline, 2022.

[D] RectanglingGAN: Deep rectangling for stitched image via image inpainting, 2024.

[E] Diff-Plugin: Revitalizing Details for Diffusion-based Low-level Tasks, 2024.

---

> ### Author Response · Authors · 2024-08-13
>
> Dear Area Chair,
>
> We write this comment in the hope that you can remind the Reviewers to reply to the Rebuttal. The time is running out, but only one Reviewer has replied now (Thanks for the timely reply from Reviewer pLdt).
>
> According to Notifications, we as authors cannot rush Reviewers to reply, which should be the responsibility of the Area Chair. We also do not know whether the Area Chair has reminded Reviewers of the deadline, because we have not received any Notification from Area Chair.
>
> Since there is only one day left and there are still three reviewers who have not replied, we hope the Area Chair will remind the reviewers of the deadline.
>
> Whether the score is changed or not, the reviewers' reply to our Rebuttal will be a good reference for our subsequent work. We believe that this is why Neurips started the discussion session. We don't think a discussion session where most people are silent is what Neurips hopes to see.

---

> > ### Comment · Area_Chair_h8Lo · 2024-08-13
> > **reminder was sent**
> >
> > Dear authors,
> >
> > The reminders were sent to individual reviewers without posting to the public. I hope that they will post soon.
> >
> > AC

---

### Decision · Program_Chairs · 2024-09-25

**Decision:**

Accept (poster)

**Comment:**

The paper proposes to unify the fusion and rectangling stages of image stitching into a single stage, formulated as an inpainting problem. With this formulation, the proposed method effectively incorporates prior knowledge within pre-trained diffusion models to address the unified problem. Initially, there were concerns about whether the inpainting formulation is a suitable solution for rectangling and fusion, as previous work provided counter-arguments. However, these arguments were made before the emergence of diffusion models, which could be a game-changer. Although the results are imperfect, the proposed method offers a novel perspective on utilizing diffusion models for image stitching. Although several other issues were raised, the rebuttal effectively addressed most. Following the discussion stage, the consensus among the reviewers is positive. Please follow the reviewers' suggestions and include the updated clarifications and experiments in the revised version.